# Rainfall threshold calculation for debris flow early warning in areas with scarcity of data

**Hua-li Pan [1, 2], Yuan-jun Jiang [1, 2, ✉], Jun Wang [3], Guo-qiang Ou [1, 2]**

✉ Corresponding author's e-mail: yuanjun.jiang.civil@gmail.com

[1] Key Laboratory of Mountain Hazards and Earth Surface Process, Chinese Academy of Sciences, Chengdu 610041, China

[2] Institute of Mountain Hazards and Environment, Chinese Academy of Sciences, Chengdu 610041, China

[3] Guangzhou Institute of Geography, Guangzhou 510070, China

**Abstract:** Debris flows are one of the natural disasters that frequently occur in mountain areas, usually accompanied by serious loss of lives and properties. One of the most used approaches to mitigate the risk associated to debris flows is the implementation of early warning systems based on well calibrated rainfall thresholds. However, many mountainous areas have little data regarding rainfall and hazards, especially in debris flow forming regions. Therefore, the traditional statistical analysis method that determines the empirical relationship between rainstorm and debris flow events cannot be effectively used to calculate reliable rainfall thresholds in these areas. After the severe Wenchuan earthquake, there were plenty of diposits deposited in the gullies which resulted in lots of debris flow events subsequently. The triggering rainfall threshold has decreased obviously. To get a reliable and accurate rainfall threshold and improve the accuracy of debris flow early warning, this paper developed a quantitative method, which is suit for debris flow triggering mechanism in meizoseismal areas, to identify rainfall threshold for debris flow early warning in areas with scarcity of data based on the initiation mechanism of hydraulic-driven debris flow. First, we studied the characteristics of the study area, including meteorology, hydrology, topography and physical characteristics of the loose solid materials. Then, the rainfall threshold was calculated by the initiation mechanism of the hydraulic debris flow. The comparison with other models and with alternate configurations demonstrates that the proposed rainfall threshold curve is a function of

the antecedent precipitation index (*API*) and 1-h rainfall. To test the proposed method, we selected the Guojuanyan gully, a typical debris flow valley that during the 2008-2013 period experienced several debris flow events and that is located in the meizoseismal areas of Wenchuan earthquake, as a case study. The comparison with other threshold models and with configurations shows that the selected approach is the most promising to be used as a starting point for further studies on debris flow early warning systems in areas with scarcity of data.

**Keywords:** Debris flow; rainfall threshold curve; rainfall threshold; areas with scarcity of data

**1 Introduction**

Debris flow is rapid, gravity-induced mass movement consisting of a mixture of water, sediment, wood and anthropogenic debris that propagate along channels incised on mountain slopes and onto debris fans (Gregoretti et al., 2016). It has been reported in over 70 countries in the world and often causes severe economic losses and human casualties, seriously retarding social and economic development (Imaizumi et al., 2006;Tecca and Genevois, 2009; Dahal et al., 2009; Liu et al., 2010; Cui et al., 2011; McCoy et al., 2012; Degetto et al., 2015; Tiranti and Deangeli, 2015; Hu et al., 2016). Rainfall is one of the main triggering factors of debris flows and is the most active factor when debris flows occur, which also determines the temporal and spatial distribution characteristics of the hazards. As one of the important and effective means of non-engineering disaster mitigation, much attention has been paid to debris flow early warning by researchers (Pan et al., 2013; Guo et al., 2013; Zhou et al., 2014; Wei et al., 2017). For rainstorm triggered debris flows, the precipitation and intensity of rainfall are the decisive factors of debris flow initiation, and a reasonable rainfall threshold target is essential to ensure the accuracy of debris flow early warning. However, if there are some extreme events occurred, such as an earthquake, the rainfall threshold of debris flow may change a lot. Tang et al. (2012) analyzed the critical rainfall of Beichuan city and found that the cumulative rainfall triggering debris flow decreased by 14.8%-22.1% when compared with the pre-earthquake period, and the critical hour rainfall decreased by 25.4%-31.6%. Chen et al. (2013)analyzed the pre- and post-earthquake critical rainfall for debris flow of Xiaogangjian gully and found that the critical rainfall for debris flow in 2011 was approximately 23% lower

than the value during the pre-earthquake period. Other researches, such as Chen et al. (2008)
and Shied et al. (2009) has reached similar conclusions that the post-earthquake critical
rainfall for debris flow is markedly lower than that of the pre-earthquake period. The
Guojuanyang gully, a small gully located in the meizoseismal areas of the big earthquake, has
no debris flows under the annual average rainfall before 2008, but it became a debris flow
gully after the earthquake under the same conditions, even the rainfall was smaller than the
annual average rainfall. These indicated that earthquakes have a big influence on debris flow
occurrence. The earthquake triggered many unstable slopes, collapses, and landslides, which
have served as the source material for debris flow and shallow landslide in the years after the
earthquake (Tang et al. 2009, 2012; Xu et al. 2012; Hu et al. 2014). Therefore, the rainfall
threshold of debris flow post-earthquake is an important and urgent issue to study for debris
flow early warning and mitigation.
As an important and effective means of disaster mitigation, debris flow early warning
have received much attention from researchers. The rainfall threshold is the core of the debris
flow early warning , on which have a great deal of researches yet (Cannon et al., 2008; Chen
and Huang 2010; Baum and Godt, 2010;Staley et al., 2013; Winter et al., 2013; Zhou and Tang,
2014; Segoni et al., 2015; Rosi et al 2015). Although the formation mechanism of debris flow
has been extensively studied, it is difficult to perform distributed physically based modeling
over large areas, mainly because the spatial variability of geotechnical parameters is very
difficult to assess (Tofani et al., 2017). Therefore, many researchers (Wilson and Joyko, 1997;
Campbell, 1975; Cheng et al., 1998) have had to determine the empirical relationship between
rainfall and debris flow events and to determine the rainfall threshold depending on the
combinations of rainfall parameters, such as antecedent rainfall, rainfall intensity, cumulative
rainfall, et al.. Takahashi (1978), Iverson (1989)and Cui (1991) predicted the formation of
debris flow based on studies ofslope stability, hydrodynamic action and the influence of pore
water pressure on the formation process of debris flow. Caine (1980) first statistically
analyzed the empirical relationship between rainfall intensity and the duration of debris flows
and shallow landslides and proposed an exponential expression( $I = 14.82D^{-0.39}$ ). Afterwards,
other researchers, such as Wieczorek (1987), Jison (1989), Hong et al. (2005), Dahal and
Hasegawa (2008), Guzzetti et al. (2008) and Saito et al. (2010), carried out further research
on the empirical relationship between rainfall intensity and the duration of debris flows,
established the empirical expression of rainfall intensity - duration ($I = D$) and proposed
debris flow prediction models. Although I-D is the most used approach, other rainfall
parameters have been considered as well for debris flow thresholds. Shied and Chen (1995)
established the critical condition of debris flow based on the relationship between cumulative
rainfall and rainfall intensity. Zhang (2014) developed a model for debris flow forecasting
based on the water-soil coupling mechanism at the watershed scale. In addition, some
researchers have highlighted the importance to find more robust hydrological bases to
empirical rainfall thresholds for landslide initiation (Bogaard et al., 2018; Canli et al., in
review; Segoni et al., 2018). When data are scarce, a robust validation of a threshold model
can be based on a quantitative comparison with alternate versions of the threshold
(Althuwaynee et al.,2015) or with thresholds calculated with completely different approaches
(Frattini et al., 2009; Lagomarsino et al., 2015). Zhenlei Wei et al. (2017) investigated a
rainfall threshold method for predicting the initiation of channelized debris flows in a small
catchment, using field measurements of rainfall and runoff data.
Overall, the studies on the rainfall threshold of debris flow can be summarized as two
methods: the demonstration method and the frequency calculated method. The
demonstration method employs statistical analysis of rainfall and debris flow data to study the
relationship between rainfall and debris flow events and to obtain the rainfall threshold curve
(Bai et al., 2008; Tian et al., 2008; Zhuang, et al., 2009). The I-D approaches would be this
kind of method. This method is relatively accurate, but it needs very rich, long-term rainfall
database and disaster information; therefore, it can be applied only to areas with a history of
long-term observations. The frequency calculated method, assuming that debris flow and
torrential rain have the same frequency, and thus, debris flow rainfall threshold can be
calculated based on the rainstorm frequency in the mountain towns where have abundant
rainfall data but lack of disaster data (Yao, 1988; Liang and Yao, 2008). Researchers have also
analyzed the relationship between debris flow occurrences and precipitation and soil moisture
content based on initial debris flow conditions (Hu and Wang, 2003). However, this approach
is rarely applied to the determination of debris flow rainfall thresholds because it needs series
of rainfall data. Pan et al. (2013) calculated the threshold rainfall for debris flow pre-warning

by calculating the critical depth of debrisflow initiation combined with the amount and regulating factors of runoff generation.

Most mountainous areas have little data regarding rainfall and hazards, especially in Western China. Neither the traditional demonstration method nor frequency calculated method can satisfy the debris flow early warning requirements in these areas. Therefore, how to calculate the rainfall threshold in these data-poor areas has become one of the most important challenges for the debris flow early warning systems. To solve this problem, this paper developed a quantitative method of calculating rainfall threshold for debris flow early warning in areas with scarcity of data based on the initiation mechanism of hydraulic-driven debris flows.

## 2 Study site

### 2.1 Location and gully characteristics of the study area

The Guojuanyan gully in Du Jiangyan city, located in the meizoseismal areas of the Wenchuan earthquake, China, was selected as the study area (Fig. 1). It is located at the Baisha River, which is the first tributary of the Minjiang River. The seismic intensity of the study area was XI, which was the maximum seismic intensity of the Wenchuan earthquake. The Shenxi Gully Earthquake Site Park is at the right side of this gully. The area extends from 31°05′27″ N to 31°05′46″ N latitude and 103°36′58″ E to 103°37′09″ E longitude, covering an area of 0.15 km² with a population of 20 inhabitants. The elevation range is from 943 m to 1222 m, the average gradient of the main channel is 270‰ (the average slope angle is 15.1°), and the length of the main channel is approximately 580m.

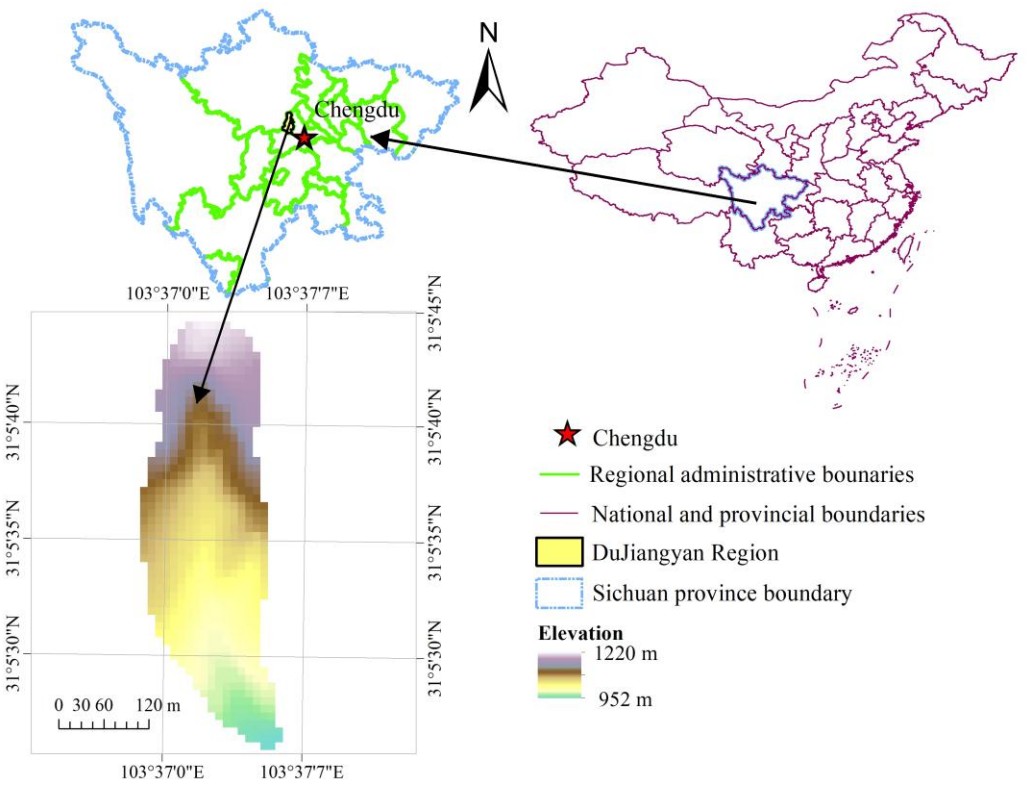


**Figure 1.** The location of the Guojuanyan gully


Geologically, the Guojuanyan gully is composed of bedrock and Quaternary strata. The
bedrock is upper Triassic Xujiahe petrofabric ($T_3x$) whose lithology is mainly sandstone;
mudstone; carbonaceous shale belonging to layered, massive structures; and semi solid-solid
petrofabric. The Quaternary strata are alluvium ($Q_4^{el+pl}$), alluvial materials ($Q_4^{pl+dl}$), landslide
accumulations and debris flow deposits ($Q_4^{sef+del}$). The thickness of the Quaternary strata
ranges from 1 m to 20 m and varies greatly. The strata profile of the Guojuanyan gully is
shown in Fig. 2.

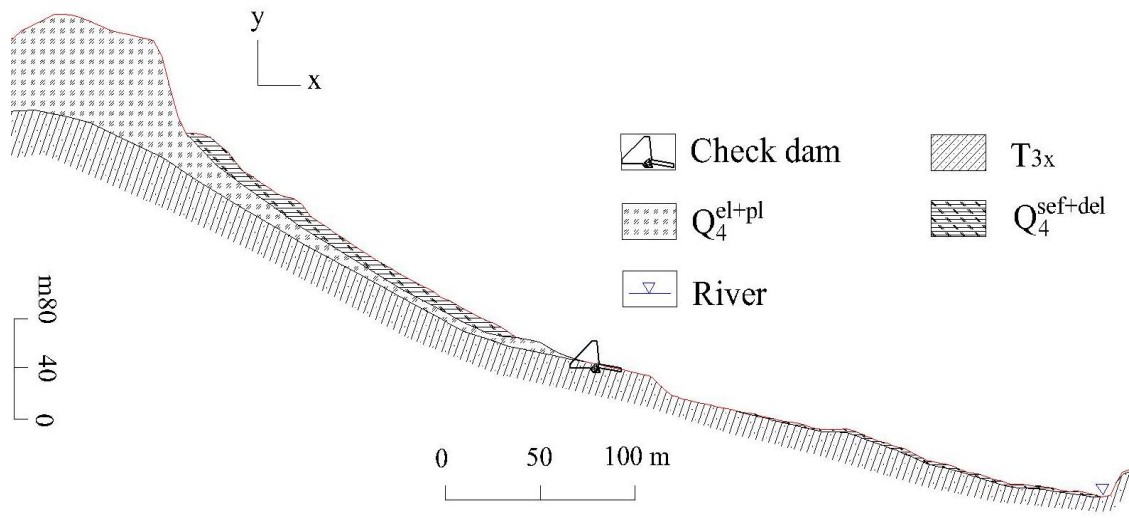

**Figure 2.** The strata profile of the Guojuanyan gully (Jun Wang et al, 2017)

Geographically, the study area belongs to the Longmenshan Mountains. The famous Longmenshan tectonic belt has a significant effect on this region, especially the Hongkou-Yinxiu fault. The study area has strong tectonic movement and strong erosion, and the main channel is "V"-shaped. The area is characterized by a rugged topography, and the main slope gradient interval of the gully is 20° to 40°, accounting for 52.38% of the entire study area.

Climatically, this area has a subtropical and humid climate, with an average annual temperature of 15.2°C and an average annual rainfall of 1200 mm (Wang et al., 2014).

**2.2 Materials and debris flow characteristics of the study area**

The Wenchuan earthquake generated a landslide in the Guojuanyan gully, leading to an abundance of loose deposits that have served as the source materials for debris flows. A comparison of the Guojuanyan gully before and after the Wenchuan earthquake is shown in Fig. 3. According to the field investigation and field tests, the landslide 3D characteristics induced by the earthquake and the infiltration characteristics of the loose materials are shown in Table 1 and Table 2 (Wang et al., 2016). They indicate that the volume of materials is more than $20 \times 10^4$ $m^3$,and the infiltration capable of the earth surface have much increased. Therefore, the trigger rainfall for debris flow has decreased greatly. The Guojuanyan gully had no debris flows before the earthquake because of the lack of loose solid materials before the earthquake; however, it became a debris flow gully after the earthquake, and debris flows occurred in the

following years (Table 3). The specific conditions of these debris flow events were collected
through field investigations and interviews. The field investigations and experiments deter-
mined that the density of the debris flow was between 1.8 and 2.1 g/cm³. Unfortunately, there
were no rainfall data before 2011, when we started field surveys in the Guojuanyan gully.

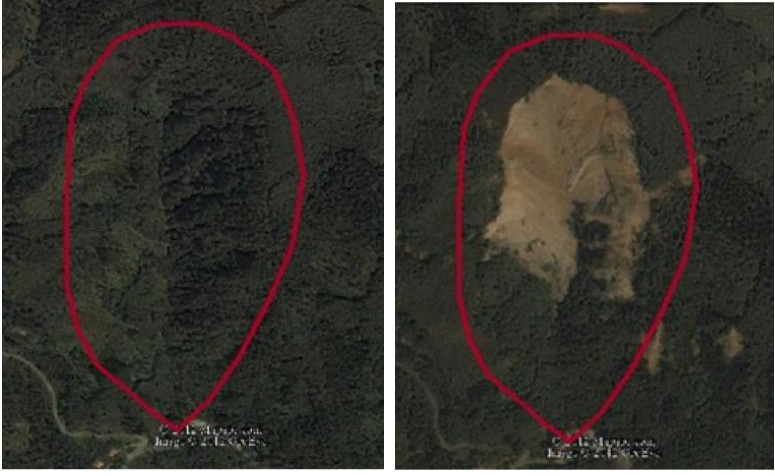


(a) 14 September, 2006     (b) 28 June, 2008

**Figure 3.** The Guojuanyan gully before (a) and after the Wenchuan earthquake (b) (from Google Earth)
**Table 1.** The landslide 3D characteristics induced by the earthquake in the study area

| Average length /m | Average width /m | Average Height /m | Average depth /m | Slope /° | Volume /×10⁴m³ |
|---|---|---|---|---|---|
| 160 | 80 | 180 | 15 | $\geqq 30$ | 20 |


**Table 2.** The infiltration characteristics of solid materials in the study area

| Infiltration curve | Infiltration rate | |
|---|---|---|
| | Initial infiltration /cm/min | Stable infiltration /cm/min |
| f= 0.6529*exp(-0.057*t) | 3.52 | 0.34 |

**Table 3.** The specific conditions of debris flow events in the Guojuanyan gully after the earthquake

| Time | Volume (10⁴ m³) | Surges | Rainfall data record |
|---|---|---|---|
| 24 September, 2008 | 0.6 | 1 | No |
| 17 July, 2009 | 0.8 | 1 | No |
| 13 August, 2010 | 4.0 | 3 | No |
| 17 August, 2010 | 0.4 | 1 | No |
| 1 July, 2011 | 0.8 | 1 | Yes |
| 17 August, 2012 | 0.7 | 1 | Yes |
| 9 July, 2013 | 0.4 | 1 | Yes |
| 26 July, 2013 | 2.0 | 2 | Yes |
| 18 July, 2014 | 1.5 | 1 | Yes |

## 2.3 Debris flow monitoring and streambed survey of the study area

After the Wenchuan earthquake, continuous field surveillance was undertaken in the study area. A debris flow monitoring system was also established in the study area. To identify the debris flow events, this monitoring system recorded stream water depth, precipitation and real-time video of the gully (Fig. 4). The water depth was measured using an ultrasonic level meter, and precipitation was recorded by a self-registering rain gauge. The real-time video was recorded onto a data logger and transmitted to the monitoring center, located in the In-stitute of Mountain Hazards and Environment, Chinese Academy of Sciences. When a rain-storm or a debris flow event occurs, the realtime data, including rainfall data, video record, and water depth data, can be observed and queried directly in the remote client computer in the monitoring center. Fig. 5 shows images taken from the recorded video. These data can be used to analyze the rainfall or other characteristics, such as the 10-min, 1- and 24-h critical rainfall. The recorded video is usually used to analyse the whole inundated process of debris flow events and to identify debris flow events as well as the data from rainfall, flow depth, and field investigation.

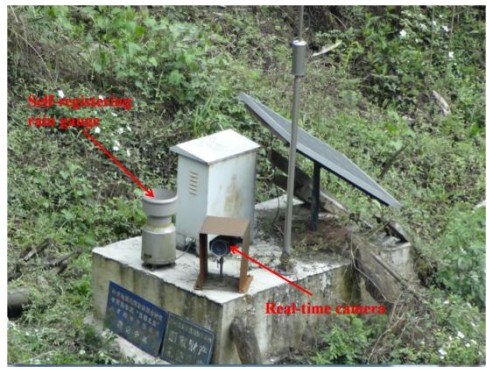 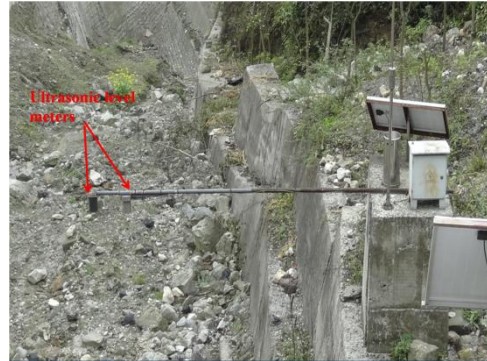

(a)Real-time camera and rain gauge      (b) Ultrasonic level meters

**Figure 4.** Debris flow monitoring system in the study area

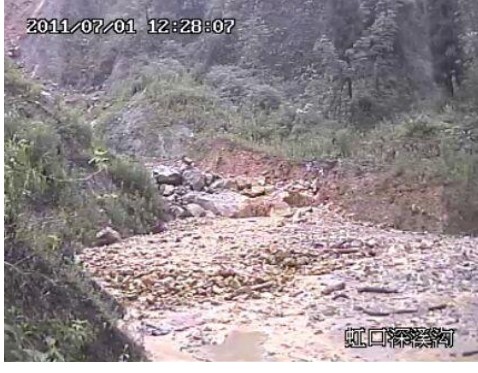 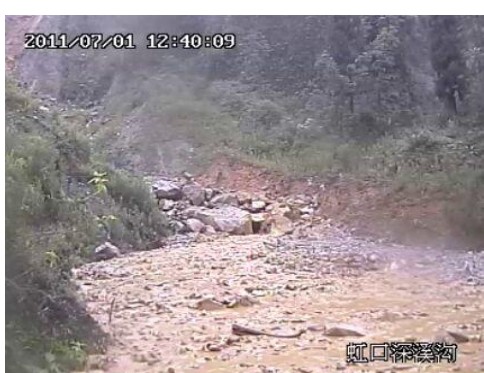

         **Figure 5.** Real-time images from video taken during the debris flow movement

**2.4 Data collection and the characteristics of rainfall**
The Wenchuan earthquake occurred in the Longmenshan tectonic belt, located on the
eastern edge of the Tibetan plateau, China, which is one of three rainstorm areas of Sichuan
Province (Longmen mountain rainstorm area, Qingyi river rainstorm area and Daba moun-
tain rainstorm area). Heavy rainstorms and extreme rainfall events occur frequently. Because
there were few data in the mountain areas, we collected the rainfall data from 1971- 2000 and
2011-2012 (from our own on-site monitoring); the characteristics of the rainfalls are as fol-
lowing:
(1) Abundant precipitation: The average annual precipitation was 1177.3 mm from 1971 to
2000, and the average monthly precipitation is shown in Fig. 6. From 1971 to 2000, the min-
imum annual precipitation of 713.5 mm occurred in 1974, and the maximum annual precipi-
tation of 1605.4 mm occurred in 1978. The total precipitation in 2012 is 1148mm, in the trend
range of the historical data.

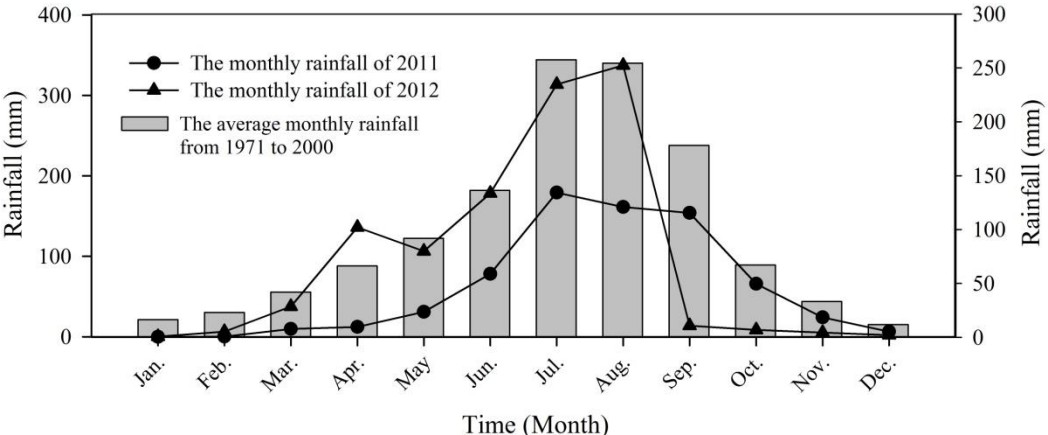


**Figure 6.** The average monthly precipitation of the Guojuanyan gully from 1971 to 2000 and the
monthly rainfall of 2011 and 2012

(2) Seasonality of the distribution of precipitation: from Fig. 6 we can observe that rain-
fall is seasonal, with approximately 80% of the total rainfall occurring during the monsoon
season (from June to September) and the other 20% in other seasons. And the laws of
monthly rainfall in 2011 and 2012 coincide to the historical data. For instance, in 2012, the
total annual rainfall in this area was approximately 1148 mm, and rainfall in the monsoon
season from June to September was 961 mm, accounting for 83.7% of the annual total.
(3) The rainfall intensity has great differences. From 1971 to 2000, the maximum month-
ly rainfall was 592.9 mm, the daily maximum rainfall was 233.8 mm, the hourly maximum
rainfall was 83.9 mm, the 10 minute maximum rainfall was 28.3 mm, and the longest contin-
uous rainfall time was 28 days.
Debris flow field monitoring data and on-site investigation data were used to identify the
debris flow events and to analyze the characteristics of the rainfall pattern and the critical
rainfall characteristics. Analyzing the typical rainfall process curves (Fig. 13), we can find that
the hourly rainfall pattern of the Guojuanyang gully is the peak pattern, displaying the single
peak and multi-peak, a characteristic of short-duration rainstorms. Through the statistical
analysis of the 10-min, 1-, and 24-h critical rainfall of debris flow events after the earthquake,
their characteristics can be obtained, as shown in Fig. 7.

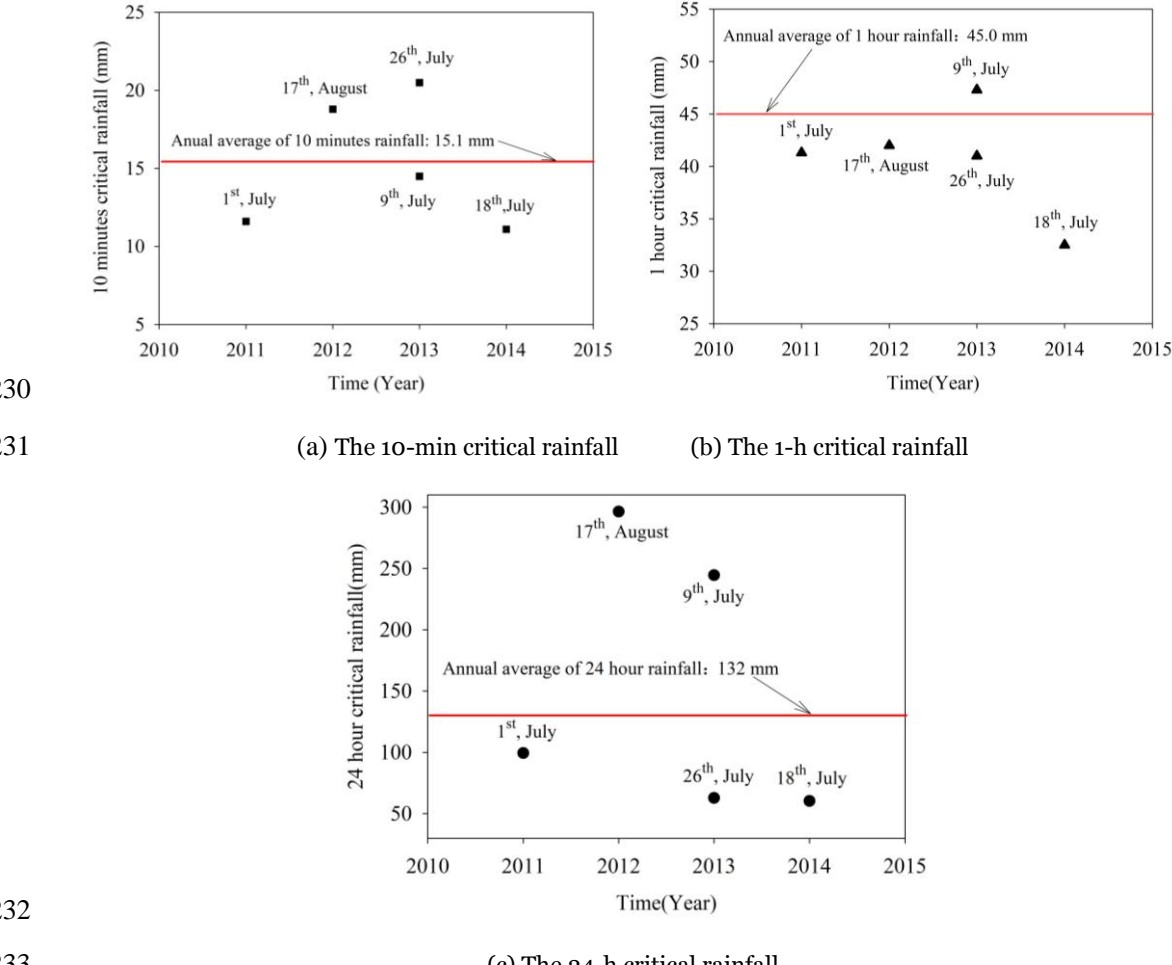


(a) The 10-min critical rainfall       (b) The 1-h critical rainfall


(c) The 24-h critical rainfall

**Figure 7.** The critical rainfall of debris flows in the Guojuanyan gully

According to the Sichuan Hydrology Record Handbook (Sichuan Water and Power De-
partment 1984), during 1940-1975, the annual average of maximum 10-min rainfall of the
study area is approximately 15.1 mm, the maximum 1-h rainfall is 45.0 mm and the annual
average of maximum 24-h rainfall is 132 mm. Fig. 7 shows that the majority of the debris flow
events in 2011-2014 occurred in a rainfall below the annual average values. This can be a con-
sequence of Wenchuan earthquake, which sensibly lowered the triggering rainfall threshold in
the test site.

## 3 Materials and methods

This study makes an attempt to analyze the trigger rainfall threshold for debris flow by
using the initiation mechanism of debris flow. Firstly, to analyze the rainfall characteristics of
the watershed by using the field monitoring data; then to calculate the runoff yield and con-
centration progress based on field observation. Additionally, the critical runoff depth to initi-
ate debris flow was calculated by the initiation mechanism with the underlying surface condi-
tion (materials, longitudinal slope, etc.) of the gully. Then, the corresponding rainfall for the
initiation of debris was back-calculated based on the stored- full runoff generation. At last,
these factors were combined to build the rainfall threshold model. This method can be applied
to the early warning system in the areas with scarcity of rainfall data.
The flow chart of the research is shown in Fig. 8.

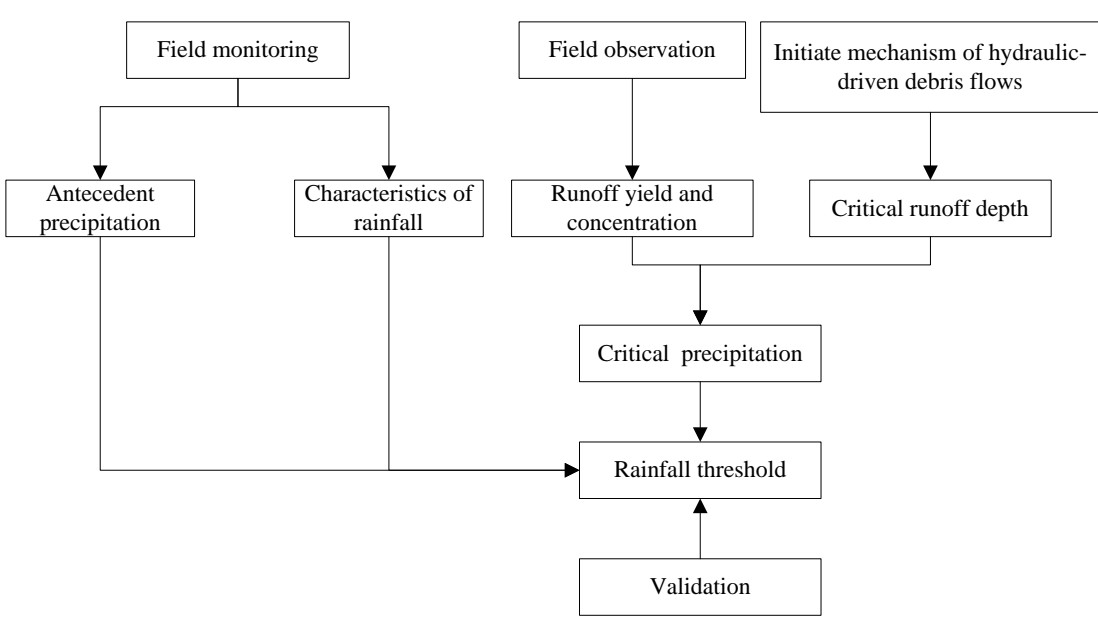

**Figure 8.** The flow chart of the research

The main influence factors for the formation of debris flow event include three parts: a
steep slope of the gully (served as potential energy condition), abundant solid materials
(source condition) and water source condition (usually is rainfall condition for rainstorm
debris flow). For rainstorm debris flow events, the precipitation and intensity of rainfall are
the decisive factors of debris flow initiation. If there is no earthquakes or other extreme events,
the topography of the gully can be considered relatively stable. In contrast, rainfall conditions
and the distribution of solid materials that determine the occurrence of debris flows can
display temporal and spatial variation within the same watershed. Therefore, it is common to
provide warning of debris flows based rainfall data after assessing the supply and distribution
of loose solid materials. In Takahashi's model, the characteristics of soil, such as the porosity
and the hydraulic conductivity of soils, are not considered, and considered the characteristic
particle size and the volume concentration of sediment; while the characteristics of
topography is mainly represented by the longitudinal slope of the gully. Furthermore, in the
stored-full runoff model, the maximum storage capacity of watershed, which mainly decided
by the porosity and permeability of the soil, may represent the characteristic of the hydraulic
conductivity of solid material to a certain extent. Therefore, this study wouldn't consider the
hydraulic conductivity any more.

## 3.1 Rainfall pattern and the spatial-temporal distribution characteristics

Mountain hazards such as debris flows are closely related to rainfall duration, rainfall
amount and rainfall pattern (Liu et al., 2009). Rainfall pattern not only affects the formation
of surface runoff but also affects the formation and development of debris flows. Different
rainfall patterns result in different soil water contents; thus, the internal structure of the soil,
stress conditions, shear resistance, slip resistance and removable thickness can vary. The ini-
tiation of a debris flow is the result of both short-duration heavy rains and the antecedent
rainfall (Cui et al., 2007; Guo et al., 2013). Many previous observational data have shown that
the initiation of a debris flow often appears at a certain time that has a high correlation with
the rainfall pattern (Rianna et al., 2014; Mohamad Ayob Mohamadi, 2015).
The precipitation characteristics not only affect the formation of runoff, also affect the
formation and development of the debris flow. Different rainfalls result in different soil water
contents, and thus the internal structure of the soil, stress conditions, corrosion resistance
and slip resistance can vary (Pan et al., 2013). Based on the rainfall characteristics, rainfall
patterns can be roughly divided into two kinds, the flat pattern and the peak pattern, as shown
in Fig. 9. If the rainfall intensity has little variation, there is no obvious peak in the whole
rainfall process; such rainfall can be described as flat pattern rainfall. If the soils characterized
by low hydraulic conductivity, this kind of rainfall can hardly trigger a debris flow separately,
and the debris flows will mainly be triggered by the great amount of effective antecedent pre-
cipitation. While if the rainfall intensity increases suddenly during a certain period of time,
the rainfall process will have an obvious peak and is termed peak pattern rainfall. If the hy-
draulic conductivity is high enough, the rainfall can totally entering the soil and mass can
move easily. These debris flows are mainly controlled by the short-duration heavy rains. Peak
pattern rainfall may have one peak or multi-peak (Pan, et al., 2013).

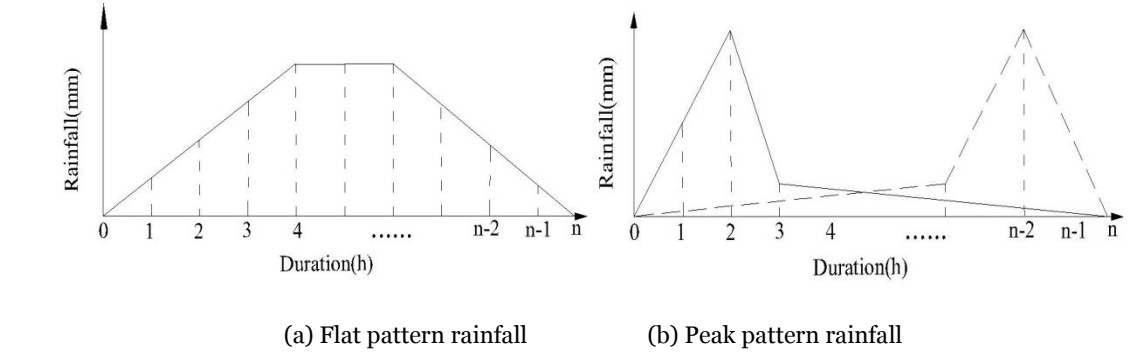


(a) Flat pattern rainfall          (b) Peak pattern rainfall

**Figure 9.** The diagram of rainfall patterns

Through analyzing the rainfall data of the Guojuanyan gully, the rainfall pattern and the
spatial-temporal distribution characteristics can be obtained.
**3.2 The calculation of the antecedent precipitation index** ( *API* )
The rainfall factor influencing debris flows consists of three parts: indirect antecedent
precipitation (IAP) (it is $P_{a0}$ in this paper), direct antecedent precipitation (DAP) (it is $R_t$ in this
paper), and triggering precipitation (TP) (it is $I_{60}$ in this paper). The relationships among them
are shown in Figure 10. Obviously, IAP increases soil moisture and decreases the soil stability,
and DAP saturates soils and thus decrease the critical condition of debris flow occurrence.
Although TP is believed to initiate debris flows directly, its contribution amounts to only 37%
of total water (Cui et al. 2007). Guo et al (2013) analyzed the rainstorms and debris flow
events during June and September in 2006 and 2008, there were 208 days with antecedent
rainfall more than 10mm, approximately 57% days of the rain season. Among them, there
were 66 days with antecedent rainfall between 10-15mm, and 1 debris flow event happened;
53 days between 15-20 mm and 4 debris flow events happened; 28 days between 20-25 mm
and 4 debris flow events happened; 30 days between 25-33 mm and 5 debris flow happened;
and 35 days more than 33mm and 9 debris flow events happened. So this group of data can
specifically illustrate the importance of the antecedent rainfall to the debris flow events.

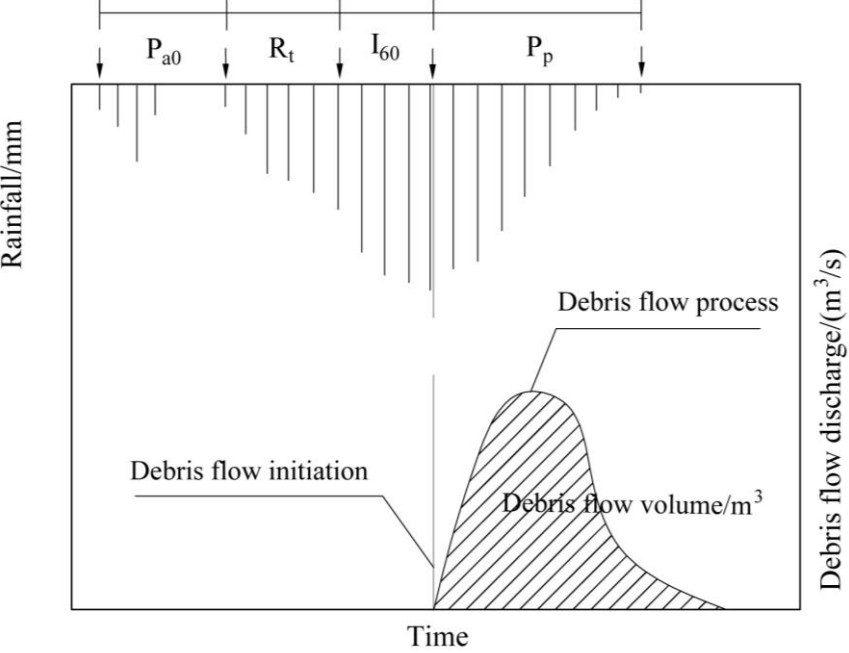


Figure 10. Rainfall index classifications
As Fig. 10 shows, take 1-h rainfall ($I_{60}$) that obtained from the observed data of the
Guojuanyan gully for the TP. The antecedent precipitation index ($API$) includes IAP and
DAP, calculated as the following expression (Zhao, 2011; Guo, 2013; Zhuang, 2015):

$$API = P_{a0} + R_t \qquad (1)$$

where $P_{a0}$ is the effective antecedent precipitation (mm) and $R_t$ is the direct antecedent precip-
itation (mm), which is the precipitation from the beginning of the rainfall that trigger debris
flow to the 1 hour before the debris flow.
It's difficult to study the influence of antecedent rainfall to debris flow as it mainly relies
on the heterogeneity of soils (strength and permeability properties), which makes it hard to
measure the moisture. Usually, the frequently used method for calculating antecedent daily
rainfall is the weighted sum equation as below (Crozier and Eyles 1980; Glade et al. 2000):
$$P_{a0} = \sum_{1}^{n} P_i \cdot K_i \qquad (2)$$

Where $P_i$ is the daily precipitation in the i-th day proceeding to the debris flow event
($1 \le i \le n$) and $K_i$ is a decay coefficient due to evaporation and geomorphological conditions
of the soil. The value of the $K$, is typically 0.8-0.9, can be determined by the test of soil mois-
ture content based on Eq.2 in the watershed. The effect of a rainfall event usually diminishes
with the time going forward. Different patterns of storm debris flow gullies require different
numbers of previous indirect rainfall days (*n*), which can be determined by the relationship
between the triggering rainfall and the antecedent rainfall of a debris flow (Pan, et al., 2013).
If the rainfall is sharp and heavy, the initiation of debris flow would mainly be determined by
DAP and TP, while the influence of the antecedent precipitation would be decreased, and vice
versa.
**3.3 The rainfall threshold curve of debris flows**
**3.3.1 The initiation mechanism of hydraulic-driven debris flows**

When the watershed hydrodynamics, which include the runoff, soil moisture content and

the discharge, reach to a certain level, the loose deposits in the channel bed will initiate
movement and the sediment concentration of the flow will increase, leading the sediment
laden flow to transform into a debris flow. The formation of this kind of debris flow is a com-
pletely hydrodynamic process. Therefore, it can be regarded as the initiation problem of de-
bris flow under hydrodynamic force. The forming process of hydraulic-driven debris flows is
shown in Fig. 10.
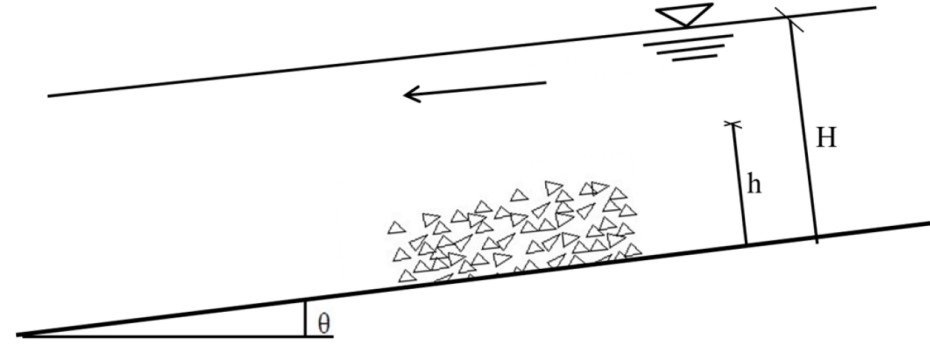

**Figure 11.** The typical debris flow initiate model

According to Takahashi's model, the critical depth for hydraulic-driven debris flows is:

$$h_0 = \left[ \frac{C_*(\sigma - \rho)\tan\phi}{\rho\tan\theta} - \frac{C_*(\sigma - \rho)}{\rho} - 1 \right] d_m \qquad (3)$$

where $C_*$ is the volume concentration obtained by experiments(0.812); $\sigma$ is the unit weight of
loose deposits (usually is 2.65 g/cm³); $\rho$ is the unit weight of water,1.0 g/cm³; $\theta$ is the chan-
nel bed slope (° ); $\phi$ is the internal friction angle (° ) and can be measured by shear tests ;
And $d_m$ is the average grain diameter (mm), which can be expressed as:

$$d_m = \frac{d_{16} + d_{50} + d_{84}}{3} \qquad (4)$$

where $d_{16}$ , $d_{50}$ and $d_{84}$ are characteristic particle sizes of the loose deposits (mm), whose
weight percentage are 16%, 50% and 84% separately.
Takahashi's model became one of the most common for the initiation of debris flow after
it was presented. A great deal of related studies was published based on Takahashi's model
later. Some discussed the laws of debris flow according to the geomorphology and the water
content (Sassa et al., 2010; Wang, 2016), while others examined the critical conditions of de-
bris flow with mechanical stability analysis (Cao et al., 2004; Jiang et al., 2017). However,
Takahashi's relation was determined for debris flow propagating over a rigid bed, hence, with
a minor effect of quasi-static actions near the bed. Lanzoni et al. (2017) slightly modified the
Takahashi's formulation of the bulk concentration, which considered the long lasting grain
interactions at the boundary between the upper, grain inertial layer and the underlying static
sediment bed, and validated the proposed formulation with a wide set of experimental data
(Takahashi, 1978, Tsubaki et al., 1983, Lanzoni, 1993, Armanini et al., 2005). The effects of
flow rheology on the basis of velocity profiles are analyzed with attention to the role of differ-
ent stress-generating mechanisms.
This study aims to the initiation of loose solid materials in the gully under surface runoff;
the interactions on the boundary are not involved. Therefore, Takahashi's model can be used
in this study.
**3.3.2 Calculation of watershed runoff yield and concentration**
The stored-full runoff, one of the modes of runoff production, is also called as the super

storage runoff. The reason of the runoff yeild is that the aeration zone and the saturation zone of the soil are both saturated. In the humid and semi humid areas where rainfall is plentiful, because of the high groundwater level and soil moisture content, when the losses of precipitation meet the plant interception and infiltration, it would not increase anymore with the rains continuous. The Guojuanyan gully is located in Du Jiangyan city, which is in a humid area. Therefore, stored-full runoff can be used to calculate the watershed runoff. That is, it can be supposed that the water storage can reach the maximum storage capacity of the watershed in each heavy rain event. Therefore, the rainfall loss in each time $I$ is the difference between the maximum water storage capacity $I_m$ and the soil moisture content before the rain $P_a$. The water balance equation of stored-full runoff is expressed as follows (Ye, et al., 1992):

$$R = P - I = P - (I_m - P_a) \tag{5}$$

where $R$ is the runoff depth (mm); $P$ is the precipitation of one rainfall (mm); $I$ is the rainfall loss (mm); $I_m$ is the watershed maximum storage capacity (mm) for a certain watershed, it is a constant for a certain watershed that can be calculated by the infiltration curve or infiltration experiment data. In this study, $I_m$ has been picked up from Handbook of rainstorm and flood in Sichuan (Sichuan Water and Power Department 1984); and $P_a$ is the antecedent precipitation index, referring to the total rainfall prior to the 1 hour peak rainfall leading to debris flow initiation.

Eq. 5 can be expressed as follows:

$$P + P_a = R + I_m \tag{6}$$

The precipitation intensity is a measure of the peak precipitation. At the same time, the duration of the peak precipitation is generally brief, lasting only up to tens of minutes. Therefore, 10-minute precipitation intensity (maximum precipitation over a 10-minute period during the rainfall event) is selected as the triggering rainfall for debris flow, which is appropriate and most representative. However, it is difficult to obtain such short-duration rainfall data in areas with scarcity of data. Therefore, in this study, $P$ and $P_a$ are replaced by $I_{60}$ (1 hour rainfall) and $API$ (the antecedent precipitation index), respectively; thus, Eq. 6 is expressed as:

$$I_{60} + API = R + I_m \tag{7}$$

In the hydrological study, the runoff depth $R$ is:
$$R=\frac{W}{1000F}=\frac{3.6\sum Q\cdot \Delta t}{F}=\frac{3.6Q}{F}$$ (8)
where $R$ is the runoff depth (m); $W$ is the total volume of runoff (m³); $F$ is the watershed area
(km²); $\Delta t$ is the duration time, in this study it is 1 hour; and $Q$ is the average flow of the water-
shed (m³/s), which can be expressed as follows:
$$Q = BVh_0$$ (9)
where $B$ is the width of the channel (m), $V$ is the average velocity (m/s) and $h_0$ is the critical
depth (m).
Eq. 7 is the expression of the rainfall threshold curve for a watershed, which can be used
for debris flow early warning. This proposed rainfall threshold curve is a function of the ante-
cedent precipitation index ($API$) and 1 hour rainfall ($I_{60}$), which is a line and a negative
slope.
**4 Results**
**4.1 The rainfall threshold curve of debris flow**
**4.1.1 The critical depth of the Guojuanyan gully**
The grain grading graph (Fig. 11) is obtained by laboratory grain size analysis experi-
ments for the loose deposits of the Guojuanyan gully. Figure 11 shows that the characteristic
particle sizes $d_{16}$, $d_{50}$, $d_{84}$ and $d_m$ are 0.18 mm, 1.9 mm, and 10.2 mm, 4.1 mm, respective-
ly. According to Eq. (1), the critical depth ($h_0$) of the Guojuanyan gully is 7.04 mm.

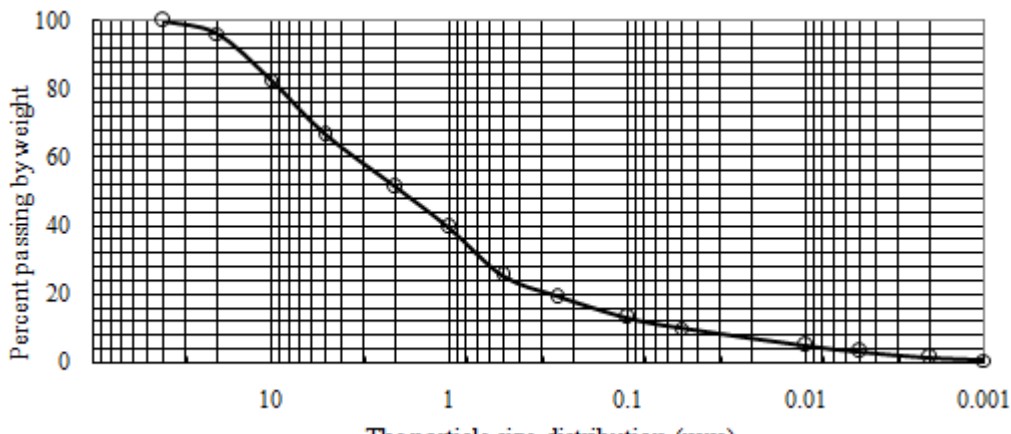

 **Figure 11.** The grain grading graph of the Guojuanyan gully

**Table 4.** Critical water depth of debris flow triggering in Guojuanyan gully

| $C_*$ | $\sigma$ (g/cm³) | $\rho$ (g/cm³) | $\tan\theta$ | $d_{16}$ (mm) | $d_{50}$ (mm) | $d_{84}$ (mm) | $d_m$ (mm) | $\phi$ (°) | $\tan\phi$ | $h_0$ (mm) |
|---|---|---|---|---|---|---|---|---|---|---|
| 0.812 | 2.67 | 1.0 | 0.333 | 0.18 | 1.9 | 10.2 | 4.1 | 21.21 | 0.388 | 7.04 |

**4.1.2 The rainfall threshold curve of debris flow**
Taking the cross-section at the outlet of the debris flow formation region as the computa-
tion object, based on the field investigations and measurements, the width of the cross-section
is 20 m, and the average velocity of debris flows which is calculated by the several debris flow
events, is 1.5m/s. Based on the Handbook of rainstorm and flood in Sichuan (Sichuan Water
and Power Department 1984), the watershed maximum storage capacity ( $I_m$ ) of the
Guojuanyan gully is 100mm. According to Eq. (5) - Eq. (7), the calculated rainfall threshold
curve of debris flow in the Guojuanyan gully is shown in Table 5.
**Table 5.** The calculated process of the rainfall threshold

| Watershed | $h_0$ (mm) | $B$ (m) | $V$ (m/s) | $Q$ (m³/s) | $\Delta t$ (h) | $F$ (km²) | $R$ (mm) | $I_m$ (mm) | $R+I_m$ (mm) |
|---|---|---|---|---|---|---|---|---|---|
| Guojuanyan | 7.04 | 20.0 | 1.5 | 0.197 | 1 | 0.11 | 6.9 | 100 | 106.9 |

From the calculated results, we can conclude the rainfall threshold of the debris flow is
$I_{60} + API = R + I_m = 106.9 \approx 107$   mm; that is, when the sum of the antecedent precipitation in-
dex ( $API$ ) and the 1 hour rainfall ( $I_{60}$ ) reaches 107 mm (early warning area), the gully may
trigger debris flow.

## 4.2 Validation of the results

### 4.2.1 The typical debris flow events in the Guojuanyan gully after earthquake

Five typical debris flow events and the corresponding rainfall processes are showed in
Figure 13. The debris flow initiation time and the rainfall, both hourly rainfall and cumulative
rainfall, have been recorded. From Fig.13, the five debris flows were triggered by torrential
rains.

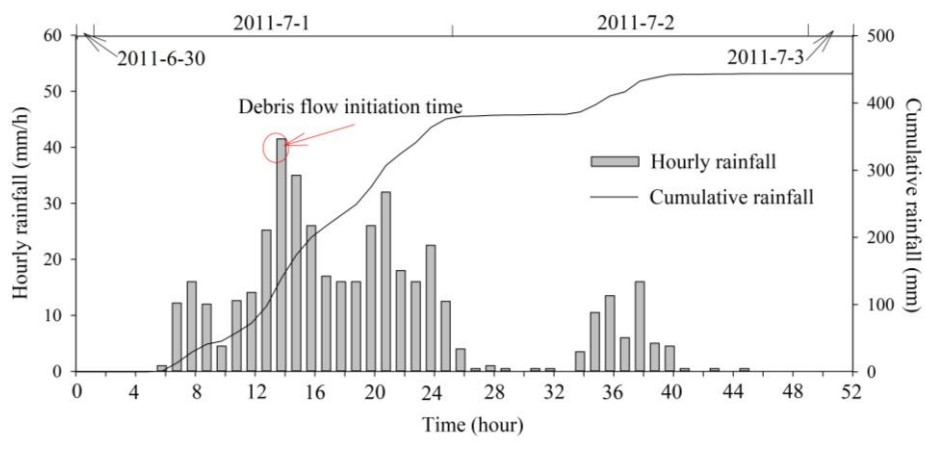


(a)

(b)

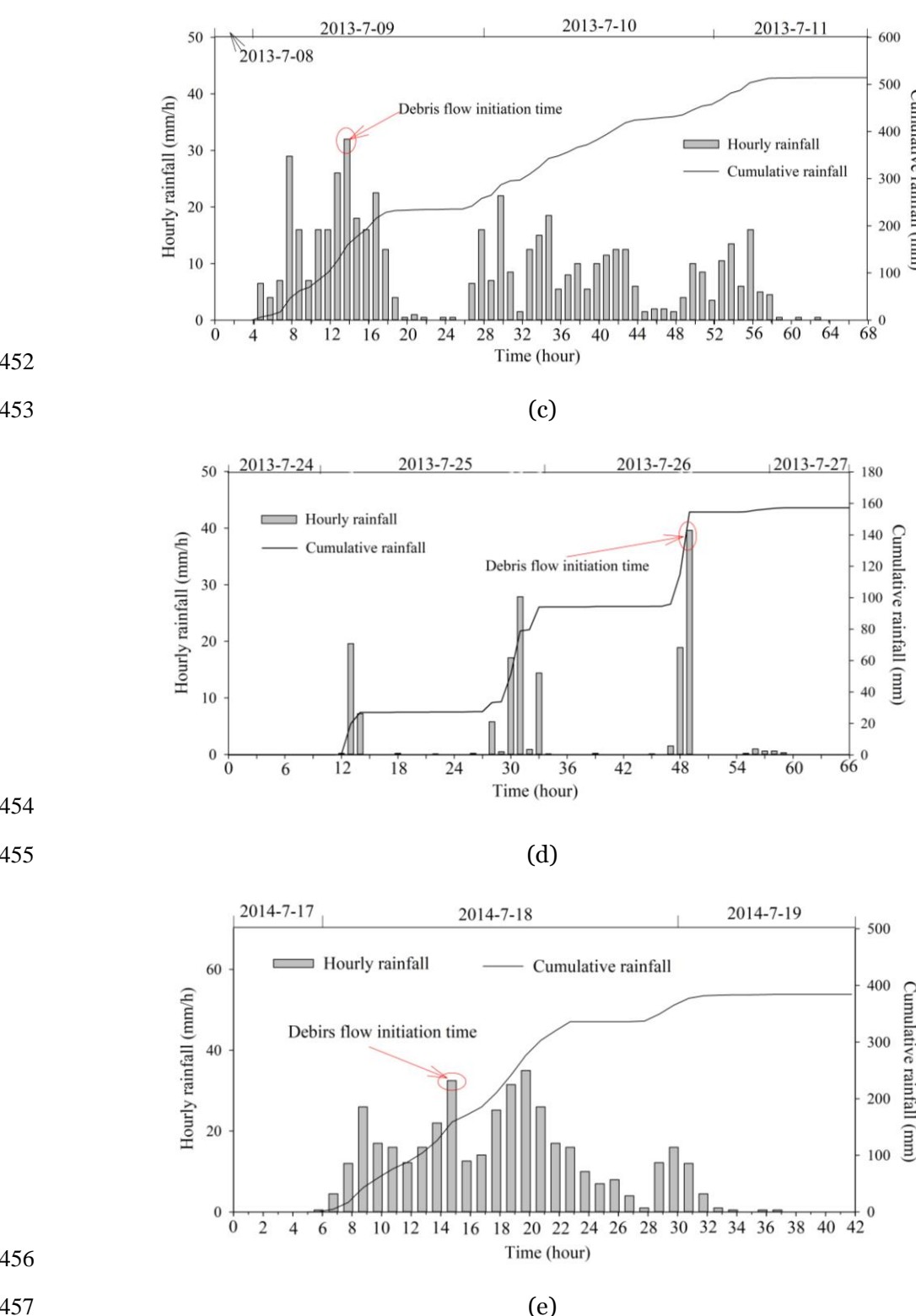


(c)

(d)

(e)
**Figure 13.** The rainfall process of debris flow events in the Guojuanyan gully from 2011 to 2014 (a, July
1, 2011; b, August 17, 2012; c, July 9, 2013; d, July 26, 2013; e, July 18, 2014)
**4.2.2 The calculation of *API* and 1-h triggering rainfall of the typical rain-**
**storms during 2010-2014**

Based on the field tests and experiences, the value of $K$ in Eq.2 is identified as 0.8 (Cui et al. 2007). To determine the numbers of previous indirect rainfall days ($n$), a comparison among 3 days, 10days, 20days and 30 days were showed in Table 6. It indicates that the value of the effective antecedent precipitations ($P_{a0}$) were increasing from 3 days to 20 days, while with the time last to 30 days, the value of $P_{a0}$ was barely changed. Therefore, it can be considered that the effect of a rainfall event usually diminished in 20 days. Hence, the numbers of previous indirect rainfall days ($n$) is identified as 20.

**Table 6.** The comparisons of $P_{a0}$ when $n$ have different values

| Time | $Pa_0$(mm) | | | |
|---|---|---|---|---|
| | n=3 | n=10 | n=20 | n=30 |
| July 1, 2011 | 3.4 | 5.2 | 9.7 | 9.7 |
| August 17, 2012 | 2.3 | 4.7 | 12.1 | 12.1 |
| July 9, 2013 | 0.8 | 2.5 | 5.7 | 5.7 |
| July 26, 2013 | 6.2 | 10.8 | 22.4 | 22.6 |
| July 18, 2014 | 0 | 6.2 | 10.7 | 10.7 |
| August 20, 2011 | 0 | 8.3 | 8.5 | 8.6 |
| September 5, 2011 | 21.3 | 45.9 | 48.7 | 48.8 |
| June 16, 2012 | 0 | 2.7 | 5.6 | 5.6 |
| August 3, 2012 | 5.6 | 6.1 | 7.5 | 7.5 |
| August 18, 2012 | 10.2 | 18.4 | 54.3 | 54.3 |
| June 18, 2013 | 0 | 2.8 | 6.2 | 6.2 |
| July 28, 2013 | 0.2 | 1.7 | 13.4 | 13.5 |
| August 6, 2013 | 0.2 | 6.6 | 12.4 | 12.4 |

Thus, the intensity of the 1-h triggering rainfall $I_{60}$ and cumulative rainfall for the typical rainstorms are shown in Table 7. In addition to the rainfall process of the 5 debris flow events (Fig. 13), some typical rainfalls whose daily rainfall were greater than 50 mm but did not trigger a debris flow were also calculated as a contrast; the greatest 1-h rainfall is considered as $I_{60}$.

**Table 7.** The data of typical rainfall in the Guojuanyan gully after the earthquake

| Time | Daily rainfall (mm) | $Pa_0$ (mm) | $R_t$ (mm) | $API$ (mm) | $I_{60}$ (mm) | $API+I_{60}$ (mm) | Location to the threshold line | Triggered debris flow |
|---|---|---|---|---|---|---|---|---|
| 1 July, 2011 | | 9.7 | 97.6 | 107.3 | 41.5 | 148.8 | Above | Yes |
| 17 August , 2012 | | 12.1 | 81.9 | 94.0 | 42.3 | 136.3 | Above | Yes |
| 9 July , 2013 | | 5.7 | 127.5 | 133.2 | 32 | 165.2 | Above | Yes |
| 26 July , 2013 | | 22.4 | 96.0 | 118.4 | 18.9 | 137.3 | Above | Yes |
| 18 July, 2014 | | 10.7 | 116.2 | 126.9 | 32.5 | 159.4 | Above | Yes |
| 20 August , 2011 | 82.8 | 8.5 | 19.0 | 27.5 | 26.8 | 54.3 | Below | No |
| 5 September , 2011 | 52.1 | 48.7 | 1.2 | 49.9 | 16.2 | 66.1 | Below | No |

| | | | | | | | |
|---|---|---|---|---|---|---|---|
| 16 June , 2012 | 55.8 | 5.6 | 6.6 | 12.2 | 27.0 | 39.2 | Below | No |
| 3 August , 2012 | 148.3 | 7.5 | 84.3 | 91.8 | 26.7 | 118.5 | Above | No |
| 18 August , 2012 | 125.7 | 54.3 | 0 | 54.3 | 65.0 | 119.3 | Above | No |
| 18 June , 2013 | 50.6 | 6.2 | 3.8 | 10.0 | 40.0 | 50.0 | Below | No |
| 28 July , 2013 | 59.4 | 13.4 | 30.0 | 43.4 | 29.4 | 72.8 | Below | No |
| 6 August , 2013 | 56.1 | 12.4 | 34.0 | 46.4 | 17.1 | 63.5 | Below | No |


The proposed rainfall threshold curve is shown in Figure 14, in which the red real line de-
fines the threshold relationship. It shows that the calculated values $I_{60} + API$ of debris flow
events in the Guojuanyan gully are all above the rainfall threshold curve, while most of the
rainstorms that did not trigger debris flow are lay below the curve. Therefore, it indicates that
the rainfall threshold curve calculated by this work is reasonable through the validation by
rainfall and hazards data of the Guojuanyan gully.

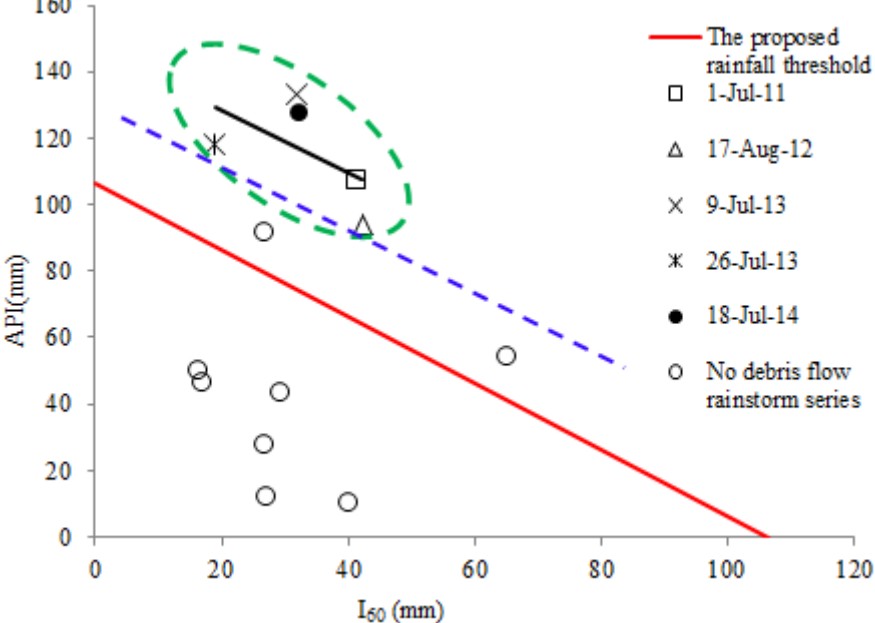


**Figure 14.** The calculated rainfall threshold curve (red real line), the trend line (black real line) of the

debris flow events and the debris flows triggering thresholds (dashed line) in Guojuanyan gully

**5 Discussions**
The trend of the debris flow events as well as the debris flow thresholds were analyzed in
Fig. 14 by using the monitoring rainfall data. A comparison between the thresholds and the
calculated threshold curve indicates that they have the same laws. Therefore, the threshold
calculated method proposed in this work is reasonable and can be used in the areas with scar-
city of data.    The proposed rainfall threshold curve is a function of the antecedent precipita-
tion index ($API$) and the 1-h rainfall ($I_{60}$), which has been validated by rainfall and hazards
data. It should be noted that the proposed approach is based on a procedure that can be ex-
ported elsewhere only if a site-specific calibration is used to develop specific thresholds for
other test sites. Therefore, the specific value of the threshold should be calculated by the initi-
ation conditions of the debris flow in specific gully.
However, this work still has two limitations. In Figure 14, there are two points above the
curve that did not trigger debris flow at all. Although we have highlighted the significance and
interconnect of antecedent rainfall, critical rainfall, 1-h triggering rainfall, as well as their ac-
curate determination before the hour of debris flow triggering, it should be noticed that the
rainfall is only the triggering factor of debris flows. A comprehensive warning system must
contain more environmental factors, such as the geologic and geomorphologic factors, the
distribution of material source. In addition, the special and complex formative environment of
debris flow after earthquake caused the rainfall threshold is much more complex and uncer-
tain. The rainfall threshold of debris flow is influenced by the antecedent precipitation index
($API$), rainfall characteristics, amount of loose deposits, channel and slope characteristics,
and so on. Therefore, we should further study the characteristics of the movable solid materi-
als, the shape of gully, and so on to modify the rainfall threshold curve. But, on the other
hand, if given the two rainstorms under the threshold, all the debris flow events points will
still locate above the threshold and there will have no missed alarms. Therefore, the threshold
established in this work is a conservative one and respect safety.
On the other hand, restricted by the limited rainfall data, this study was validated by only
5 debris flow events. Another limitation of this work is that the approach proposed in this
study hasn't been validated by other gullies except the Guojuanyan gully so far. Figure 13 and
Figure 14 indicated that the only 5 debris flow events all triggered by the rainfalls with
high-intensity and short-duration. In the future, the value of the curve should be further vali-
dated and continuously corrected with more rainfall and disaster data in later years.
**6 Conclusions**
(1) In the Wenchuan earthquake affected areas, loose deposits are widely distributed,
causing dramatic changes on the environmental development for the occurrence of debris
flow; thus, the debris flow occurrence increased dramatically in the subsequent years. The

characteristics of the 10-min, 1-h and 24-h critical rainfalls were represented based on a comprehensive analysis of limited rainfall and hazards data. The statistical results show that the 10-min and 1-h critical rainfalls of different debris flow events have minor differences; however, the 24 hour critical rainfalls vary widely. The 10-min and 1-h critical rainfalls have a notably higher correlation with debris flow occurrences than to the 24-h critical rainfalls.

(2) The rainfall pattern of the Guojuanyan gully is the peak pattern, both single peak and multi-peak. The antecedent precipitation index ( $API$ ) was fully explored by the antecedent effective rainfall and triggering rainfall.

(3) As an important and effective means of debris flow early warning and mitigation, the rainfall threshold of debris flow was determined in this paper, and a new method to calculate the rainfall threshold is put forward. Firstly, the rainfall characteristics, hydrological characteristics, and some other topography conditions were analyzed. Then, the critical water depth for the initiation of debris flows is calculated according to the topography conditions and physical characteristics of the loose solid materials. Finally, according to the initiation mechanism of hydraulic-driven debris flow, combined with the runoff yield and concentration laws of the watershed, this study promoted a new method to calculate the debris flow rainfall threshold. At last, the hydrological condition for the initiation of a debris flow is the result of both short-duration heavy rains ( $I_{60}$ ) and the antecedent precipitation index ( $API$ ). The proposed approach resolves the problem of debris flow early warning in areas with scarcity data, can be used to establish warning systems of debris flows for similar catchments in areas with scarcity data although it still need further modification. This study provides a new thinking for the debris flow early warning in the mountain areas.

## Acknowledgments

This paper was supported by the CRSRI Open Research Program (Program No. CKWV2015229/KY), CAS Pioneer Hundred Talents Program, the fund of Institute of Mountain Hazard and Environment (No.sds-135-1701), and National Nature Science Foundation of China (51679229). It was also supported by Youth Innovation Promotion Association of the Chinese Academy of Sciences (2018405).

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
