# Peer review of "Rainfall threshold calculation for debris flow early"

_Natural Hazards and Earth System Sciences, 2017_

## Short Comment (SC1) · 10 Oct 2017

dear authors

We will like to add a comment on this technical note, referring to the section 2.2.1 "The initiation mechanism of hydraulic-driven debris flows". In the manuscript it is adopted the formula of the sediment concentration suggested by Takahashi (1977) (reported in eq. (1) of the paper) for describing the initiation mechanism of hydraulic-driven debris flows. The Takahashi relation was determined for stony debris flows propagating over a rigid bed and, hence, with a minor effects of quasi-static actions near the bed. In order to obtain a correct estimate of the bulk concentration, the long lasting grain interactions at the boundary between the upper, grain inertial layer and the underlying

static sediment bed should be accounted for. A recently published paper Lanzoni et al. [2017], slightly modified the mentioned Takahashi formulation, and validate the proposed formulation with a wide dateset of experimental data (Takahashi [1978], Tsubaki et al. [1983], Lanzoni [1993], Armanini et al. [2005]). In particular, in the absence of an appreciable excess basal pore fluid pressure the friction coefficient (reported as alpha in the present manuscript) is replaced by the quasi-static friction angle ðĺIJŚqs associated with prolonged interparticle contacts. Adopting the formulation suggested by Lanzoni et al. [2017] the difierence between the measured values of C and those resulting from the new formulation are mostly contained in the ±10% range.

Laura Stancanelli and Carlo Gregoretti

Armanini, A., H.Capart, L.Fraccarollo, and M.Larcher (2005),Rheological-stratification in experimental free-surface flows of granular-liquid mixtures, J.FluidMech.,532,269–319,doi:10.1017/S0022112005004283. Lanzoni, S. (1993), Meccanica di miscugli solido-liquido in regime granulo-inerziale, PhD thesis, Univ. Of Padova (inItalian), Padua, Italy. Lanzoni, S., C. Gregoretti, and L. M. Stancanelli (2017), Coarse-grained debris flow dynamics on erodible beds, J. Geophys. Res. Earth Surf., 122, doi:10.1002/2016JF004046. Takahashi, T. (1978), Mechanical characteristics of debris flow, J. Hydraul. Div. ASCE, 104(8), 1153–1169. Tsubaki, T., H. Hashimoto, and T.Suetsugi (1983), Interparticle stresses and characteristics of debris flows, Hydrosci. Hydrraul. Eng., 1, 67–82.

---

## Referee Comment (RC1) · Anonymous Referee #1 · 11 Oct 2017

General Comments The paper explores a very interesting topic: the occurrence of debris-flows after earthquakes producing huge amounts of loose deposits remained in the channels and on the slopes. To this aim, a simple physical model is proposed; it is calibrated and "validated" on few cases available on the area. The cited literature is adequate and the approach could be reliable. However, data characterizing mechanical, rheological and hydraulic behavior of the soil are not properly displayed. Moreover, the reliability of the physical approach for such cases is not properly substantiated. In particular, the choice of accounting for antecedent precipitations avoiding to adopt usual I-D approaches should be justified. In this perspective, the paper should be substantially improved according my view. Under such constraints, it could be reconsidered only after performing major revisions Furthermore, specific comments and requests for

clarifications/modifications are reported below:

Abstract

L28: please amend "scarcity" for "scaricty"

1 Introduction

L32-80: probably, reorganizing the first part of abstract could help readability; my proposal is first introducing debris flow and rainfall thresholds, after debris flow post earthquake and associated thresholds with the focus on debris flows post 2008 earthquake

L72: please amend "triggeringdebris" in "triggering debris"; please check the entire Manuscripts where several typos are recognized

L82: please stress the deep uncertainties affecting "frequency calculated method"

2 Materials and methods

L106-108: please check font size

L109-110: what do you mean for "The characteristics of rainfall in the watershed were analyzed firstly by the field survey" (in this sense, also further details for figure 1 should be provided)

L124-126: grain-size distribution regulates hydraulic properties and then duration and intensity of rainfalls triggering the event; please introduce such elements about it

L129: please cite as "Rianna et al., 2014"

L130-138: the assumed link between debris flow initiation and rainfall pattern should be deepened; as reported in previous item, hydraulic properties of soils involved regulate what type of rainfalls can generate or not phenomena. As general rule, the higher the conductivity, the larger the influence of short heavy rainfall events able to totally entering the soil; on the other side, for soils characterized by low hydraulic conductivity, cumulative values on longer time spans are relevant for mass movements. L146:

please move the Figure 3 below under the related text.

L148-156: please stress the constraints associated to such assumptions

L161: avoid the term "density" for soil particles; "unit weight of soil" could be preferable

L162: please check font size

L172: avoid the term "density" for soil particles; "unit weight of soil" could be preferable

L172-174: please specify if such parameters can be assumed constant or featured for such soils; in this case, please move in "Case Study" section

L176-177: please provide further details or brief definitions for d16,d50,d84

L180: please specify what you intend for "stored-full runoff"

L190: please confirm that Im is roughly represented by porosity for soil depth

L196: why is 1h assumed as reference duration?

L202: what do you intend for "computational step"?

L204: how do you define such parameters?

3 Case study

L218-219: please check the number of inhabitants

L254: you could consider the table a simple list of events occurred; frequency is not calculated

L263: please define "abnormal"; in this perspective, the rainfall threshold could be used to define rainfalls of interest

L265: please correct "monitroring"

L283: please correct as "Figure 9"

L282: you could report also reference percentiles of PDF (e.g. 25 and 75) in order to evaluate if 2011 and 2012 trends are included in range

L297-300: for debris flow, a graph similar to Figure 9 for monthly average maximum daily precipitation could be very useful; in this regard, to maintain consistency, you should use 1971-2000 time span

L301-310: information about hydraulic conductivity of involved soils is crucial to understand what could be the duration of interest; also for rainfall patterns reported in Figure 10, reporting hourly rainfall values could be interesting

Figure 11: please provide further details about annual average data; of course, you calculate only on wet fraction; what is the threshold for discriminating rainfall event? E.g. 1mm/d

L320-321: please you confirm that the data reported in line in figure 11 are related to average values and not to average of maximum yearly data?

L333-338: an evaluation of hydraulic behavior is crucial; as you report short term durations are crucial. Are you sure that antecedent precipitations could play a relevant role for triggering events?

L343-351: the sentences could be moved in "Introduction" part

4 Results

L358: please check the number of equation

L359: please report on y-axis that the graph reports "Percent passing by weight"

L365: please specify in which ways the value about velocity is retrieved

L367: please specify on what soil depth you evaluate Im

L377-387: the formula is not clear; please provide further details; indeed, it is not clear why you sum rainfalls (Rt) with effective rainfalls. Moreover, K parameter should be

not related to atmospheric conditions but to geomorphological conditions regulating the "detection" time of water in the soil depth of interest (e.g. hydraulic conditions, bottom conditions, slope angle). Moreover, it could take into account the effect of evapotranspiration losses reducing the amount of soil water content . For very coarse soil, K could be very low. An interesting work about such parameter is carried out by Baum & Godt (2010) (DOI10.1007/s10346-009-0177-0) and cited works.

L396-397: the issue related to antecedent conditions is widely debated in literature; in this perspective several elements concur and then further details about involved soil are required

Table 4: it provides several information already available in Table 1; please merge the two ones

Figure 14: please provide information about why the reliability of I-D rainfall thresholds accounting for only "triggering" event has not been assessed.

[Figure]

---

## Referee Comment (RC2) · Anonymous Referee #2 · 17 Oct 2017

General comments: The paper deals with an interesting topic, which is completely within the scope of the special issue. The study area is of particular interest given the amount of loose sediments that become available after the Wenchuan earthquake. Thus, the authors propose a quantitative method to identify the critical rainfall threshold for the triggering of debris flows in a data-poor area. This is an important contribution considering that in mountain areas the availability of data is often very limited and one of the major problems regarding the development of hazards studies. However, given the debris flows initiation mechanism (surface runoff erosion) the use of the API index should be better argued. For instance, if I understood it well the authors considered the cumulative precipitation of 20 days plus the 1-hour rainfall for the triggering of debris flows. Again, this must be deeply discussed given the debris flows initiation mechanism. Therefore, I think the manuscript could be accepted after major revisions.

Furthermore, some specific comments are listed below:

Regarding the structure of the manuscript, I would suggest placing the section "3.1 Location and gully characteristics of the study area" after the "1. Introduction" and before the "2. Materials and methods".

Page 2, Line 48-49: Please, check the sentence because is not clear

Page 3, Line 58-59: The references should be chronologically displayed

Page 3, Line 67: Please, check how to cite the authors (and also along the manuscript)

Page 4, Line 88-91: Please, check the sentence

Page 4, Line 109-110: Please, explain how this was done

Page 4, Line 113-114: Please, explain why did the authors assumed a saturated condition to explain the debris flows initiated by runoff?

Page 5, Line 126-127: Please, provide some references that support this sentence

Page 5-6, Line 132-134: When you mention "the great amount of antecedent precipitation" you should clarify the temporal resolution

Page 7, Line 164-167: Please, provide references

Page 9, Line 221: Please, indicate the average slope angle of the main channel in degrees

Page 11, Line 247: Please, standardize the name of the gully along the manuscript. Sometimes is written as Guojuanyan gully, others as Guo Juanyan gully

Page 13, Line 281: In which way is evaluated the spatial variability of rainfall?

Page 17, Line 348: Replace "was present" with "become available"

[Figure]

Page 17, Line 358: Please, check the equation number

Page 17, Line 361: Please, standardize the units used in Table 2 and Equation 3

Page 19, Line 391: Please, explain how equation 12 can be used to estimate the amount of solid material

Page 23, Line 441-443: Please, check the sentence

Page 23, Line 447: Please, refer which other factors should be addressed

Finally, I suggest a rereading of the text in order to correct some minor mistakes.

---

## Referee Comment (RC3) · Anonymous Referee #3 · 31 Oct 2017

The manuscript proposes development a "quantitative method to identify rainfall threshold for debris flow early warning in data-poor areas based on the initiation mechanism of hydraulic-driven debris flow". Therefore, cannot be considered a classical debris flow mechanism, where the triggering starts on slopes with high declivity, generally from the generalized shallow landslides (models of the unsaturated soil mechanics). Unlike this, the methodological proposal of the manuscript involves modeling with physical characteristics of the loose solid materials (landslide triggered by earthquake - loose deposits that have served as the source materials for debris flows) using the equations (3) and (4) – Takahashi's model. This issue is very important and should be highlighted (emphasized) in the manuscript, mainly because the rainfall thresholds obtained in this paper cannot be generalized and used to classical debris flow's early warning systems

or, at best, used with reservation. In general, the manuscript needs to be more concise and written better. Additionally, in my understanding and opinion, some inconsistent scientific aspects of key parts of the manuscript significantly compromise their acceptance and publication, in the mode as they are. The main scientific inconsistencies of the manuscript are listed below: 3.4 Data collection and the characteristics of rainfall – in this point, the characteristics of the pattern rainfall need to be better explained scientifically, for example, as from others rainfall indexes (accumulated of 48h, 72h, 96h, etc.). In addition, some pattern rainfall indexes analyzed (lines 282 to 300) correspond to previous periods (1971 to 2000 and 1957 to 2008) to the occurrence of the debris flows events (2008 to 2014). In the case of a have information about the pattern rainfall from the debris flows events occurrence period, it is consider fundamental to analyze in detail the rainfall indexes for this period, that is, from 2008 to 2014 (take as an example the information in Figure10 – page 15). 4.1.1 The critical depth of the Guojuanyan gully – the equation (1) used for calculate the critical depth (line 358, page 17) are wrong. The correct equations are (3) and (4). 4.1.2 The rainfall threshold curve of debris flow – in the lines 368 to 369, "….rainfall threshold curve of debris flow in the Guojuanyan gully is shown in Table 3", was used which equation to calculate the threshold curve? 4.2.2 The rainstorm and debris flow events in the Guojuanyan gully during 2010-2014 Analyzing the data of the Figures 13 (a, b, c, d and e), it is observed that the triggering rainfall of debris flow events are situated well above (136 to 165 mm) of the established rainfall threshold (107 mm). The data of the Figure 14 corroborates with this statement. Additionally, two points of debris flow no occurrence were verified above of the rainfall threshold curve. Therefore, the authors' assertion (lines 433 to 437) does not match the results presented and will have to be re-analyzed. 6 Conclusions - The statements contained in the paragraph between the lines 481 to 483 need to represent better the results presented in Figures 13 and 14, this is, the rainfall threshold curve proposed should be used with caution, because it contains relevant uncertainties due to the scarcity of data.

The following are some comments aimed at improving and clarifying some points of the

manuscript: Line 101 - ". . ..method nor frequency. . ..." change to ". . . . . .method for fre-quency . . .." Line 125 - ". . .., corrosion resistance,. . .." the correct meaning is not ". . .., shear resistance,. . .."? Lines 246 to 248 - "The Guojuanyan gully had no debris flows before the earthquake; however, it became a debris flow gully after the earthquake, and debris flows occurred in the following years (Table 1)". This does not seem obvious, because before there was no material deposited! Lines 249/250 - ". . ..density of the debris flow was between 1.8 and 2.1 g/cm3. . ." the correct meaning is not ". . ..density of the soil was between 1.8 and 2.1 g/cm3,. . .."? Line 265 - ". . .., monitroring center,. . .." change to ". . . . . .monitoring center . . .." Line 321 - ". . ..obsevation. . ...." change to ". . . . . .observation. . .." Line 327 - ". . ..maxmum. . ..." change to ". . . . . .maximum. . .." Fig-ure 13 (e) – reform the label "debirs flow" Figure 13 – standardize the figure's legend

---

## Editor Comment (EC1) · S. Segoni (Editor) · 18 Nov 2017

Dear Authors, this comment is to invite you to start working on the revision of the manuscript. As you can see, the 3 referees' reports and the interactive comment pointed out several issues that need to be properly addressed. During the revision of the manuscript, I recommend you to take in great consideration ALL comments received. The revised version of the manuscript will be sent again to referees for a second round of review.

I suggest you to start working on the revision before the end of the open discussion, if you want to save some time and to speed up the review process.

Regards,

[Figure]

S.

---

## Author Comment (AC1) · 23 Nov 2017

Dear Editor Segoni,

Thanks for your comments. We have read through all the comments proposed by the three referees as well as the interactive comment in the discussion process. We would give our appreciation to all the reviewers for their very constructive and orientive suggestions those are really the points need to be improved through a major revision. As requested we are starting the revision immediately, and we expect to submit the revised manuscript with concern of all the comments as soon as possbile without any compromise of the quality.

[Figure]

2017-333, 2017.

---

## Author Comment (AC2) · 29 Nov 2017

Dear Gregoretti, Thanks for your comment on our manuscript. It's great pleasure that we can read the latest research report from you. Actually you have done an effective job on the initiation of debris flow in an erodible gully. We will take fully consideration on your results in our study to calculated the rainfall threshold. We will clarify it in our revised paper.

---

## Author Comment (AC3) · 29 Nov 2017

Dear sir, Thanks very much for your elaborate efforts on our paper and your advices are extremely important to improve the manuscript. We just want to illustrate one thing that as you mentioned in the comment, the I-D model is one of the most popular approaches to account for antecedent prcipitations in geohazards early warning researches. However, this method needs a great deal of rainfall and disaster data while many mountainous areas have little data regarding rainfall and hazards. This paper is just want to establish a method to calculate the rainfall threshold for debris flow in such areas with scarcity of data. Of course we will further clarify it in the appropriate part of the manuscript. Thanks again for your useful and specific comments, we will carefully revise the manuscript accordingly.

---

## Author Comment (AC4) · 29 Nov 2017

Dear referee, Many thanks for your comments and advices on our manuscript. Actually your suggestions are constructive and we will reorganize the structure of the article so that to make it to be more concise and clear. The very specific comments will also be considered thoroughly during the revision of the paper.

---

## Author Comment (AC5) · 29 Nov 2017

Dear referee, Thanks a lot for your comments on the "Technical notes: Rainfall threshold calculation for debris flow early warning in areas with scarcity of data" as well as your positive evaluation about the research significance of our study. As you mentioned that maybe the API index we used in this maunuscript procuce ambiguity, we will further explain and define it in our revised paper. Fourthermore, we wil clarify the antecedent effective rainfall (reported as cumulative precipitation of 20 days in the present paper) and the stimulate precipitation of debris flow (1-hour rainfall now). In one word, we truly appreciate the thoughtful comments from you and we will revise the manuscript carefully according to your advices as well as all of the specific comments.

---

## Author Response (AR1)

Dear Samuele Segoni,

Thank you for the chance that you kindly gave us about our manuscript entitled "**A**
**method of rainfall threshold calculation for debris flow early warning in da-**
**ta-poor areas—a case study in Guojuanyan gully, Sichuan Province, China**"
(No.nhess-2017-333). We truly appreciate all of the thoughtful comments from you
and Referees as well as the interactive comment, and we have now revised our manu-
script accordingly with a list of changes detailed below.

**Response to Editor:**

*1) Please carefully check the reference list and rewrite it according to the style*

*of NHESS journal, where needed.*

We have checked the references.

*2) Answering to referee comment #1 you state that "I-D model is one of the most*

*popular approaches to account for antecedent prcipitations in geohazards". I*

*disagree, as the ID approach is mostly used to account for peak intensity precip-*

*itation and it is not effective to account for antecedent precipitation. For this*

*reason, ID thresholds are used for debris flows and shallow landslides, which*

*are usually triggered by short and intense precipitation and in which antecedent*

*precipitation does not necessarily play a decisive role.*

We truly agree with you. I-D approach is mostly used to analyze the relationship be-
tween peak intensity precipitation and the hazards. We made a mistake by careless
and we wanted to say that I-D model is one of the most popular approaches to calcu-
late threshold of geological hazards.

*3) Answers to referee comments are very generic. The Authors should put better*

*efforts in revising the manuscript and in submitting a detailed list of*

*point-to-point replies and amendments to the text.*

Thanks very much for your kindly remind. We have made a careful revision and made
a point-to point replies to the text.

*4) There was a discussion among the Editorial board of NHESS and it was*

*stressed that "technical notes" are not an accepted manuscript type. Therefore, I*

*ask you to change your title in "Rainfall threshold calculation for debris flow*

*early warning in areas with scarcity of data" and to better stress the general*

*scientific outcomes of your work.*

This has been changed as you suggested.

*5) After performing all amendments to the text, please spend the due time to*

*check the text accurately for typos and for revising the English.*

We have thorouly checked the whole manuscript again.

**Response to Short comment (SC1):**

*In the manuscript it is adopted the formula of the sediment concentration sug-*

*gested by Takahashi (1977) (reported in eq. (1) of the paper) for describing the*

*initiation mechanism of hydraulic-driven debris flows. The Takahashi relation*

*was determined for stony debris ïn˙Cows propagating ˙ over a rigid bed and,*

*hence, with a minor effects of quasi-static actions near the bed. In order to ob-*

*tain a correct estimate of the bulk concentration, the long lasting grain interac-*

*tions at the boundary between the upper, grain inertial layer and the underlying*

*C1 NHESSD Interactive comment Printer-friendly version Discussion paper*

*static sediment bed should be accounted for. A recently published paper Lanzoni*

*et al. [2017], slightly modified the mentioned Takahashi formulation, and vali-*

*date the proposed formulation with a wide dateset of experimental data.*

Thank your very much for your kindly discussion. We have the honor to read the pa-
per you ciated carefully and find that you have done a very good job on the dynamics
of coarse-grained debris flow dynamics. A remarkable collapse of the dimentionless
profiles is obtained by scaling the debris flow velocity with the ruanoff velocity, and a
power law characterization is proposed following a heuristic approach. The effects of
flow rheology on the basis of velocity profiles are analyzed with attention to the role
of different stress-generating mechanisms. Especially the work on the dynamic simi-
larity is very important for it's one of the critical problems the debris flow subject
grappling with. We have sited your study in the development and applications of
Takahashi's model (Line 331-337). Our study aims to the initiation of loose solid ma-
terials in the gully under surface runoff; the interactions on the boundary are not in-
volved. Therefore, Takahashi's model can be used in this study. And this is also an
attempt to calculate the rainfall thershold in area with scarcity data; there are lots of
further works to do to. We'll try to consider your method and results in our work in
future. Thanks again for your thinking and comment.

**Response to Anonymous Referee #1:**

*1) General comments:*

*...However, data characterizing mechanical, rheological and hydraulic behavior*

*of the soil are not properly displayed. Moreover, the reliability of the physical*

*approach for such cases is not properly substantiated.*

The mothod used in this study is mainly focus on the Takahashi's model which con-
sidered the characteristic particle size and the volume concentration of sediment. And
the hydraulic conductivity of solid material would represented by the maximum
strorage capacity of watershed in the stored-full runoff model. We have made a fur-
ther illustration in the manuscript in "Materals and methods" section (Line 262-276).

*In particular, the choice of accounting for antecedent precipitations avoiding to*

*adopt usual I-D approaches should be justified.*

The I-D approaches would be demonstration method. This method is relatively accurate, but it needs very rich, long-term rainfall sequence data and disaster information; therefore, it can be applied only to areas with a history of long-term observations. This study is mainly focus on the area with scarcity data; therfore; it can't calculate the rainfall threshold by I-D approaches.

*2) Abstract*

*L28: please amend "scarcity" for "scaricty"*

It has been amended yet.

*3) Introduction:*

*L32-80: probably, reorganizing the first part of abstract could help readability;*

*my proposal is first introducing debris flow and rainfall thresholds, after debris*

*flow post earthquake and associated thresholds with the focus on debris flows*

*post 2008 earthquake*

The abtract has been rewritten according your advice.

*L32-80: please amend "triggeringdebris" in "triggering debris"; please check*

*the entire Manuscripts where several typos are recognized*

We checked the whole paper thoroughly and amended the type mistakes.

*L82: please stress the deep uncertainties affecting "frequency calculated method"*

Because the frequency calculated method also needs series of rainfall data, in the areas with scarcity of data can't use it (Line 99-100).

*4) Materials and methods:*
*L106-108: please check font size*

This has been checked.

*L109-110: what do you mean for "The characteristics of rainfall in the watershed were analyzed firstly by the field survey" (in this sense, also further details*

*for figure 1 should be provided)*

This sentence has been changed into "Firstly, to analyze the rainfall characteristics of the watershed by the field monitoring as well as record data if there is any; then to calculate the runoff yield and concentration based on field observation." (Line 249-251) and the figure (Figure 8 at present) has been changed accordingly.

*L124-126: grain-size distribution regulates hydraulic properties and then duration and intensity of rainfalls triggering the event; please introduce such elements about it*

The main influnce factors for the formation of debris flow event include the geomorphology of gully, characteristics of solid materials and high-intensity rainfall event. We have added some illustrations in the paper (Line 262-276).

*L129: please cite as "Rianna et al., 2014"*

This has been amended yet.

*L130-138: the assumed link between debris flow initiation and rainfall pattern should be deepened; as reported in previous item, hydraulic properties of soils involved regulate what type of rainfalls can generate or not phenomena. As general rule, the higher the conductivity, the larger the influence of short heavy rainfall events able to totally entering the soil; on the other side, for soils characterized by low hydraulic conductivity, cumulative values on longer time spans are relevant for mass movements.*

As metntioned in previous question, this paper put characteristics of materials and geomorphology of gully as backgroud data; hence these two were talked about little. We have rewritten this paragraph to clarify it. (Line 262-276)

*L146: please move the Figure 3 below under the related text*

This has been amended yet.

*L148-156: please stress the constraints associated to such assumptions*

The whole part had been rewitten already. Especially, the constraints of the assumption of Takahashi's model were explained in the last paragraph of this part (Line 326-337).

*L161: avoid the term "density" for soil particles; "unit weight of soil" could be preferable*

This has been amended as you suggested.

*L162: please check font size*

The whoe manuscript has been checked thoroughly and the spelling and format errors were corrected.

*L172: avoid the term "density" for soil particles; "unit weight of soil" could be preferable*

This has been amended as you suggested.

*L172-174: please specify if such parameters can be assumed constant or fea-*

*tured for such soils; in this case, please move in "Case Study" section*

In Takahashi's model, the volume concentration, the unit weight of loose deposits and
the unit weight of water are usually unchanged while the channel bed slope and the
internal friciton angle are characterized by different soils. In this part, the manuscript
is mainly introducing the methdology used in this study. The perticular values of these
parameters in this paper are showed carefully in "Case Study" section.
*L176-177: please provide further details or brief definitions for d16, d50, d84*

We added detail illustration under the Eq. (2) (Line 324-325).

*L180: please specify what you intend for "stored-full runoff"*

The stored-full runoff, one of the modes of runoff production, is also called as the su-
per storage runoff. The reason of the runoff yeild is that the aeration zone and the sa-
truration zone of the soil are saturated by rainfall. In the humid and semi humid areas
where rainfall is plentful, because of the high groundwater level and soil moisture
content, the loss of precipitation is no longer increased with the rains continue, after
meet plant interception and infiltration, which produces a wide range of surface runoff
(Line 342-347).
*L190: please confirm that Im is roughly represented by porosity for soil depth*

No, it can't represented by porosity for soil depth. $I_m$ is the maximum water storage
capacity for a specific watershed, it is a constant for a certain watershed that can be
calculated by the infiltration curve or infiltration experiment data.

*L196: why is 1h assumed as reference duration?*

The precipitation intensity is a measure of the peak precipitation. At the same time,
the duration of the peak precipitation is generally brief, lasting only up to tens of
minutes. Therefore, 10-minute precipitation intensity (maximum precipitation over a
10-minute period during the rainfall event) is selected as the stimulating rainfall for
debris flow, which is appropriate and most representative. However, it is difficult to
obtain such short-duration rainfall data in areas with scarcity of data which is just our
research range. Therefore, 1h is assumed as reference duration in this study (Line
365-370).

*L202: what do you intend for "computational step"?*

This has been changed into "$\Delta t$ is the duration time, in this study it is 1 hour" (Line
377).

*L204: how do you define such parameters?*

$Q$ is the average flow of the watershed, $B$ is the width of the channel, $V$ is the average
velocity and $h_0$ is the critical depth (Line 377-381).

*5) Case study*
*L218-219: please check the number of inhabitants*

Yes, the Guojuanyan gully is very small and there is only 20 inhabitants living in this
area.

*L254: you could consider the table a simple list of events occurred; frequency is*
*not calculated*

We have merged table 1 and table 4 into one table as you suggested in the later.

*L263: please define "abnormal"; in this perspective, the rainfall threshold could*
*be used to define rainfalls of interest*

This sentence has been changed into "When a rainstorm or a debris flow event oc-
curs,…." (Line 172-173).

*L265: please correct "monitroring"*

This has been corrected.

*L283: please correct as "Figure 9"*

This has been corrected.

*L282: you could report also reference percentiles of PDF (e.g. 25 and 75) in or-*
*der to evaluate if 2011 and 2012 trends are included in range*

Sorry we don't understand what the PDF means is. However, we analyzed the rainfall
laws in latest years and it has the same regular pattern with the historical data. (Line
203-204, Figure 6). Actually, the laws of rainfall don't change as it still in the same
rainstorm belt which would not be influenced by the earthquake.

*L297-300: for debris flow, a graph similar to Figure 9 for monthly average*
*maximum daily precipitation could be very useful; in this regard, to maintain*
*consistency, you should use 1971-2000 time span*

This has been amended as you suggested.

*L301-310: information about hydraulic conductivity of involved soils is crucial*
*to understand what could be the duration of interest; also for rainfall patterns*
*reported in Figure 10, reporting hourly rainfall values could be interesting*

Of course the hydraulic conductivity of involed soils is important. However, the
mothod used in this study is mainly focus on the Takahashi's model which considered
the characteristic particle size and the volume concentration of sediment. And the hy-
draulic conductivity of solid material would represented by the maximum strorage
capacity of watershed in the stored-full runoff model. We have made a further illus-
tration in the manuscript in "Materals and methods" section (Line 259-273). And as
you suggested, the rainfall patterns is replaced by hourly rainfall values which were showed in Figure 13 (Line 214-216).

*Figure 11: please provide further details about annual average data; of course,*

*you calculate only on wet fraction; what is the threshold for discriminating*

*rainfall event? E.g. 1mm/d*

This figure has been redrawn. It mainly calculated the critical rainfall events which
had a large precipitation.

*L320-321: please you confirm that the data reported in line in figure 11 are re-*

*lated to average values and not to average of maximum yearly data?*

Yes, it's an average of maximum yearly data and we had corrceted this (Line 226,
Line 229 and Line 235).

*L333-338: an evaluation of hydraulic behavior is crucial; as you report short*

*term durations are crucial. Are you sure that antecedent precipitations could*

*play a relevant role for triggering events?*

According to the previous studies, debris flow initiated is the result of the short dura-
tion rainfall (10-min rainfall, 1-h rainfall for example) and the effective antecedent
precipitations (Cui et al., 2007; Zhao, 2011; Guo, 2013; Zhuang, 2015).

*L343-351: the sentences could be moved in "Introduction" part*

We have moved the sentences to "Introduction" part (Line 47-56).

*6) Results*
*L358: please check the number of equation*

This has been corrected.

*L359: please report on y-axis that the graph reports "Percent passing by*

*weight"*

This has been corrected.

*L365: please specify in which ways the value about velocity is retrieved*

The average velocity of debris flows is calculated by the several debris flow events
occurred in this gully (Line 399-400).

*L367: please specify on what soil depth you evaluate Im*

$I_m$ is the maximum water storage capacity for a specific watershed, it is a constant for
a certain watershed that can be calculated by the infiltration curve or infiltration ex-
periment data.

*L377-387: the formula is not clear; please provide further details; indeed, it is*

*not clear why you sum rainfalls (Rt) with effective rainfalls. Moreover, K param-*

*eter should be not related to atmospheric conditions but to geomorphological*

*conditions regulating the "detection" time of water in the soil depth of interest*

*(e.g. hydraulic conditions, bottom conditions, slope angle). Moreover, it could*

*take into account the effect of evapotranspiration losses reducing the amount of*

*soil water content . For very coarse soil, K could be very low. An interesting*

*work about such parameter is carried out by Baum & Godt (2010)*

*(DOI10.1007/s10346-009-0177-0) and cited works.*

We have rewritten the whole part of 4.2.1 and added a rainfall index classification figure to illustrate the equations and parameters (Line 411-433, Figure 12).

*L396-397: the issue related to antecedent conditions is widely debated in litera-*

*ture; in this perspective several elements concur and then further details about*

*involved soil are required*

According to the previous studies, debris flow initiated is the result of the short dura-tion rainfall (10-min rainfall, 1-h rainfall for example) and the effective antecedent precipitations (Cui et al., 2007; Zhao, 2011; Guo, 2013; Zhuang, 2015).

*Table 4: it provides several information already available in Table 1; please*

*merge the two ones*

We have merged table 1 and table 4 into one.

*Figure 14: please provide information about why the reliability of I-D rainfall*

*thresholds accounting for only "triggering" event has not been assessed.*

I-D approaches belong to demonstration method which is the most accurate method to calculate the debris flow thershold. Howere, it needs plenty of disaster data as well as correspongding rainfall data to statistic the laws between debris flow initiation and the characteristics of rainfall. In areas with scarcity of data, actually almost areas in mountainous are this situation, there is few hazard data and rainfall data. Therefore, the I-D approaches can't satisfy the early warning of debris flow. In fact, this is the consideration of our study, to propose a new thinking for the debris flow early warn-ing in the areas with scarcity of data. We clarified this view in the "Introduction" sec-tion (Line 91-106).

**Response to Anonymous Referee #2:**

*General comments:*

*However, given the debris flows initiation mechanism (surface runoff erosion)*

*the use of the API index should be better argued. For instance, if I understood it*

*well the authors considered the cumulative precipitation of 20 days plus the 1-hour rainfall for the triggering of debris flows. Again, this must be deeply discussed given the debris flows initiation mechanism.*

According to the previous studies, debris flow initiated is the result of the short duration rainfall (10-min rainfall, 1-h rainfall for example) and the effective antecedent precipitations (Cui et al., 2007; Zhao, 2011; Guo, 2013; Zhuang, 2015). The precipitation intensity is a measure of the peak precipitation. At the same time, the duration of the peak precipitation is generally brief, lasting only up to tens of minutes. Therefore, 10-minute precipitation intensity (maximum precipitation over a 10-minute period during the rainfall event) is selected as the stimulating rainfall for debris flow, which is appropriate and most representative. However, it is difficult to obtain such short-duration rainfall data in areas with scarcity of data which is just our research range. Therefore, this study considered the effective antecedent precipitation of 20 days plus 1-h rainfall for the triggering of debris flow event.

*Regarding the structure of the manuscript, I would suggest placing the section "3.1 Location and gully characteristics of the study area" after the "1. Introduction" and before the "2. Materials and methods".*

We have changed order of section 2 and section 3, and now it is "2 Study site" and "3 Materials and methods".

*Page 2, Line 48-49: Please, check the sentence because is not clear*

This sentence has been checked (Line 59-61).

*Page 3, Line 58-59: The references should be chronologically displayed*

This has been corrected also along the whole manuscript.

*Page 3, Line 67: Please, check how to cite the authors (and also along the manuscript)*

This has been checked.

*Page 4, Line 88-91: Please, check the sentence*

This has been checked (Line 99-100).

*Page 4, Line 109-110: Please, explain how this was done*

Firstly, to analyze the rainfall characteristics of the watershed by the field monitoring as well as record data if there is any; then to calculate the runoff yield and concentration progress based on field observation. Additionally, the critical runoff depth to initiate debris flow was calculated by the initiation mechanism with the underlying surface condition (materials, longitudinal slope, etc.) of the gully (Line 249-255).

*Page 4, Line 113-114: Please, explain why did the authors assumed a saturated*

*condition to explain the debris flows initiated by runoff?*

The method in this study mainly based on the stored-full runoff generation because
the study site is in a humid area (Line 253-255, Line 342-353).

*Page 5, Line 126-127: Please, provide some references that support this sen-*
*tence*

Some references have been added (Line 284).

*Page 5-6, Line 132-134: When you mention "the great amount of antecedent*
*precipitation" you should clarify the temporal resolution*

This has been changed (Line 293-296).

*Page 7, Line 164-167: Please, provide references*

Some references have been added (Line330-337).

*Page 9, Line 221: Please, indicate the average slope angle of the main channel*
*in degrees*

The average slope angle has been added (Line127).

*Page 11, Line 247: Please, standardize the name of the gully along the manu-*
*script. Sometimes is written as Guojuanyan gully, others as Guo Juanyan gully*

The name of the gully has been unified as "Guojuanyan gully".

*Page 13, Line 281: In which way is evaluated the spatial variability of rainfall?*

Actually this study didn't analyze the spatial variability of rainfall. The sentence has
been changed as "The characteristics of the rainfall are as following" (Line191-192).

*Page 17, Line 348: Replace "was present" with "become available"*

This has been corrected.

*Page 17, Line 358: Please, check the equation number*

This has been corrected.

*Page 17, Line 361: Please, standardize the units used in Table 2 and Equation 3*

This has been corrected.

*Page 19, Line 391: Please, explain how equation 12 can be used to estimate the*
*amount of solid material*

I'm sorry, there was a mistake. It should be "Eq.9 can be used to estimate the mosis-
ture content of solid material prior to the debris flow" (Line 434-435).

*Page 23, Line 441-443: Please, check the sentence*

The sentence has been corrected.

*Page 23, Line 447: Please, refer which other factors should be addressed*

The other factors means the factors mentioned before except the rainfall characteris-
tics that accounting for in this study. We added some further illustrations in the man-
uscript (Line 478-479).

*Finally, I suggest a rereading of the text in order to correct some minor mis-*
*takes.*

As your nice suggestion, we have checked up the whole manuscript thoroughly and
corrected some spelling and format mistakes.
**Response to Anonymous Referee #3:**
*1) General comments:*
*Unlike this, the methodological proposal of the manuscript involves modeling*
*with physical characteristics of the loose solid materials (landslide triggered by*
*earthquake - loose deposits that have served as the source materials for debris*
*flows) using the equations (3) and (4) – Takahashi's model. This issue is very*
*important and should be highlighted (emphasized) in the manuscript, mainly be-*
*cause the rainfall thresholds obtained in this paper cannot be generalized and*
*used to classical debris flow's early warning systems or, at best, used with res-*
*ervation. In general, the manuscript needs to be more concise and written better.*

We added some illustrations about the applicative conditions of the method proposed
in this manuscript. Addtionally, made some discussions in the "Discussion" section.

*3.4 Data collection and the characteristics of rainfall – in this point, the charac-*
*teristics of the pattern rainfall need to be better explained scientifically, for ex-*
*ample, as from others rainfall indexes (accumulated of 48h, 72h, 96h, etc.). In*
*addition, some pattern rainfall indexes analyzed (lines 282 to 300) correspond to*
*previous periods (1971 to 2000 and 1957 to 2008) to the occurrence of the de-*
*bris flows events (2008 to 2014). In the case of a have information about the*
*pattern rainfall from the debris flows events occurrence period, it is consider*

*fundamental to analyze in detail the rainfall indexes for this period, that is, from*

*2008 to 2014 (take as an example the information in Figure10 – page 15).*

We added some more illustrations about the rainfall indexes in the manuscript (Line 189- 192, Line 196-197, Line 203-204). However, as our on-site monitoring system usually affected by the bad weather or some other reasons, we only have the ho-lonomic data of 2011 and 2012. Hence we only analyzed the yearly rainfall character-istics of these two years. As the laws of rainstorm are mainly based on the location in where the rainstorm area is, and the laws of the rainfall in 2011 and 2012 coincide to the historical data. Hence, we think this analysis can satisfy our research. In addition, as it is generally recognized that debris flow is usually triggered by short-duration rainstorms, we just analyzed the 10-min, 1-h and 24-h rainfall indexes. And the Figure 7 corroborates with this statement.

*4.1.1 The critical depth of the Guojuanyan gully – the equation (1) used for cal-culate the critical depth (line 358, page 17) are wrong. The correct equations are (3) and (4).*

This has been corrected.

*4.1.2 The rainfall threshold curve of debris flow – in the lines 368 to 369,*

*". . ..rainfall threshold curve of debris flow in the Guojuanyan gully is shown in*

*Table 3", was used which equation to calculate the threshold curve?*

The equation has been illustrated above the table (Line 402-403).

*4.2.2 The rainstorm and debris flow events in the Guojuanyan gully during*

*2010-2014 Analyzing the data of the Figures 13 (a, b, c, d and e), it is observed*

*that the triggering rainfall of debris flow events are situated well above (136 to*

*165 mm) of the established rainfall threshold (107 mm). The data of the Figure*

*14 corroborates with this statement. Additionally, two points of debris flow no*

*occurrence were verified above of the rainfall threshold curve. Therefore, the*

*authors' assertion (lines 433 to 437) does not match the results presented and*

*will have to be re-analyzed.*

The antecedent precipitation index (*API*) in the manuscript includes two parts: the ef-fective antecedent precipitation and the direcet antecedent precipitation, which is the precipitation from the beginning of the rainfall that trigger debris flow to the 1 hour before the debris flow. And the $I_{60}$ in the threshold curve is the precipitation 1 hour before the time debris flow occurred. The relationship of the rainfall indexes is shown in Figure 12. The rainfall indexes of debris flow events happened in Guojuanyan gully mentioned in figure 13 were shown in Table 4. Although the total precipitations are between 136 to 165mm, the $I_{60}$ varies from 18.9 to 42.3 mm. It is much smaller than the threshold (107 mm). It should plus the *API* value to situate above the threshold curve to trigger a debris flow. To validate the result, we added some typical raifalls whose daily rainfall were greater than 50 mm but didn't trigger a debris flow to Figure 14. All of the debris flow event's point are lay above the curve and most of the rainstorms tha didn't trigger debris flow are lay below the curve. It indicates that the proposed method is reasonalbe. However, the triggering factors for a debris flow are very complex and uncertain. Not only the factors mentioned in this study, the *API* and *I₆₀*, but also the amount of loose deposits, channel and slope characteristics, and et al. would affect the initiation of debris flow. Hence, we should further study the characteristics of the movable solid materials, the shape of gully, and so on. Maybe this is the main reason of the two points lay above the curve but didn't trigger a debris flow. We discussed this in the "Discussion" section (Line 471-479).

*6 Conclusions - The statements contained in the paragraph between the lines 481 to 483 need to represent better the results presented in Figures 13 and 14, this is, the rainfall threshold curve proposed should be used with caution, because it contains relevant uncertainties due to the scarcity of data.*

The sentences have been reorganized and modified (Line 515-519).

*Line 101 - ". . ...method nor frequency. . ..." change to ". . .. .method for frequency . . .."*

The sentence is right. It means both the traditional demonstraion method and frequency calculated method can't satisfy the debris flow early warning requirements in the areas with scarcity data.

*Line 125 - ". . .., corrosion resistance,. . .." the correct meaning is not ". . .., shear resistance,. . .."?*

This has been corrected.

*Lines 246 to 248 - "The Guojuanyan gully had no debris flows before the earthquake; however, it became a debris flow gully after the earthquake, and debris flows occurred in the following years (Table 1)". This does not seem obvious, because before there was no material deposited!*

Yes, you are right. Because there was little loose solid materials in the gully before the earthquake, there was no debris flow at all. We added a detail illustraion in the manuscript to make is much more clear (Line 152-154).

*Lines 249/250 - ". . ..density of the debris flow was between 1.8 and 2.1 g/cm3. . ." the correct meaning is not ". . ..density of the soil was between 1.8 and 2.1 g/cm3,. . .."?*

It's the debris flows' density.

*Line 265 - ". . .., monitroring center,. . .." change to ". . .. . .monitoring cen-*
*ter . . .."*
This has been corrected.

*Line 321 - ". . ..obsevation. . ..." change to ". . .. . .observation. . .."*
This has been corrected.

*Line 327 - ". . ..maxmum. . ..." change to ". . .. . .maximum. . .."*
This has been corrected.

*Figure 13 (e) – reform the label "debirs flow" Figure 13 – standardize the fig-*
*ure's legend*
This has been checked.

We wish that with the above revisions made, our manuscript can now be accepted for
publication on *Nature Hazards and Earth System Sciences* soon. Please do not hesi-
tate to contact me if you have any additional questions or comments.
Looking forward to hearing from you.
Regards
JIANG Yuanjun

[revised manuscript text omitted]

---

## Author Response (AR2)

Dear Samuele Segoni,
Thank you again for the chance that you kindly gave us about our manuscript entitled
"**Rainfall threshold calculation for debris flow early warning in areas with scar-**
**city of data**" (No. NHESS-2017-333). We truly appreciate all of the thoughtful com-
ments from you and Referees, and we have now revised our manuscript accordingly
with a list of changes detailed below.
**Response to Anonymous Referee #1:**
*1) General comments:*
*…The main is surely the choice of proxies regulating the occurrence of landslide*
*phenomena: the coupled adoption of 1h and 20 days appears worthy of deep ex-*
*planation and proper sensitivity analysis; indeed, the "triggering" rainfall event*
*is typical of very steep, shallow, coarse grained layers that should not be affected*
*by cumulative values over three weeks.*

Indeed, this study is mainly focus on the rainfall threshold for debris flow initiation,
and not for the landslide. The selection of 1-h and the significance role of the ante-
cedent rainfall have been further clarified in corresponding parts of the manuscript,
not only in "2.4 data collection and the characteristics of rainfall" (Line 253-258),
"4.2.1 the calculation of the antecedent precipitation index (API)" (Line 433-441;
Line 458-466), but also in the discussion section (Line 515-528).
*2) Abstract: it should be more concise avoiding some details, for example, about*
*slope of the curve.*

The abstract has been rewritten according to your advice.
*3) L16: please amend "earthquake"; also in this version, several typos are rec-*
*ognizable*

We checked the whole paper thoroughly and amended the type mistakes.
*4) L41-55: it represents a deepening about the case study that could be moved*
*after completing the introduction of general features. The two main topics cov-*
*ered by Introduction should not be mixed: 1) general features about debris flow;*
*2) variation of critical rainfalls after earthquake; in this perspective, the text*
*should be reorganized.*

This part has been reorganized as you suggested. The general influence of rainfall has been added to the introduction first, then the variation characteristics after earthquake has been discussed (Line 41-61).

*5) L225-226: please report the reference time over which annual average of maximum 10-min rainfall is computed. The paragraph "2.2" should report details about grain size distribution of involved soils or, hopefully, on hydraulic properties: only, through these data, we can evaluate if duration for rainfall thresholds are reliable*

The reference time over which annual average of maximum 10-min rainfall, 1-h rainfall as well as the 24-h rainfall are computed is from 1940- 1975 by read the Sichuan Hydrology Record Handbook carefully (Line 234-248).

*6) L237-245: according the paragraph, 1 hour could be not the trivial option: only an occurrence is registered beyond the threshold; please comment it*

Actually, the 10-min rainfall intensity is the most appropriate index for early warning of debris flow, which is most representative and has minor error. However, it is difficult to obtain such short-duration rainfall data in actual debris flow gullies because long-term rainfall monitoring system do not exist in most debris flow basins especially in areas with scarcity of data. We further illustrate this in the manuscript (Line 253-258).

*7) L248-249: "to analyze the rainfall characteristics of the watershed by the field monitoring as well as record data there is any" on my view, the sentence remains not so clear; please try an attempt to make it more readable*

This has been amended yet.

*8) L260-263: the three factors are not strictly comparable: the first two ones concern geomorphological features and then landslide susceptibility while the third one is related to occurrence of these phenomena and then associated hazard*

The main influence factors for the formation of debris flow include steep longitudinal slope of the gully, source condition and water source condition are well accepted by researchers. However, we rewrote the sentence to make it much more clear(Line

277-280).

*9) L272-273: "the maximum storage capacity of watershed can represent the characteristic of the hydraulic conductivity of solid material": it should be clarified why a storage capacity could represent a proxy for hydraulic conductivity*

This part has been rewritten yet (Line 290-292).

*10) L285-298: the description should be clarified: what do you intend for low hydraulic conductivity? Indeed, for low hydraulic conductivity different dynamics arise and cumulative values on very long time spans have to be accounted for. The single event is not an effective proxy for these soils.*

Indeed, this part is mainly focus on the rainfall patterns. The introduction about the hydraulic conductivity is just served as a background which has no influences on the equations or the method proposed in this study (Line 304-317).

*11) L317: does the value refer to investigated soils? Is it referred to soil grain matrix?*

Yes, the value of the volume concentration refers to investigated soils in Guojuanyan gully, and it is referred to soil grain matrix.

*12) L322-324: it could be interesting to report the entire grain size distribution of investigated soils at this point (after, Figure 11)*

Actually, this is the methodology part, and mainly introduce the research thinking and the formula maybe used in this research. That is to say it is a general introduction here. Therefor, we don't think it is necessary to report a specific grain size in this part. Indeed, for the study site, Guojuanyan gully, the grain size distribution of the soils is shown in Figure 11.

*13) L340-346: the paragraph should be improved; different terms should be introduced as they refer to specific literature field; the entire dynamics regulating "stored-full runoff"*

We have introduced the "stored-full runoff" carefully, which is common used in the humid and semi humid areas in China to analyze the runoff yield mechanism (Line

359-369).

*14) L368-369: it represents the crucial point of the research; the assumption is not trivial and should properly justified; why soils affected by 1 hour heavy rainfall should be influenced in terms of slope stability by antecedent precipitations?*

The initiation of debris flow is influenced both by the antecedent precipitation and triggering precipitation. The significance role of the antecedent rainfall has been further clarified in corresponding parts of the manuscript, not only in "2.4 data collection and the characteristics of rainfall" (Line 253-258), "4.2.1 the calculation of the antecedent precipitation index (API)" (Line 433-441; Line 458-466), but also in the discussion section (Line 515-528).

*15) Table 2: it is usual reporting the angle and not directly their tangent; please, if possible, correct it*

This has been amended, we have added a new column and filled wit the angle of the longitudinal slope.

*16) L409-410: please introduce and explain the main terms reported in the text (e.g. the differences between indirect and direct antecedent precipitation)*

This has been explained in the manuscript (Line 429-432).

*17) L433-434: the period of 20 days could result quite arbitrary*

This has been illustrated in the manuscript (Line 459-467; Line 482-485).

*18) Figure 13: in the graphs cumulative values and not intensity are reported; for 1 hour the values are comparable but in this case you should report as measurements unity: mm/h. Moreover, you could report only time on the x-axis and indications about days as graph label.*

Indeed, the x-axis is the duration time of the rainfall, while the left-axis is the 1-h rainfall while the right-axis is the cumulative rainfall during the whole rainfall pro- cess.

*19) Figure 14: the threshold is located quite below the rainfall histories inducing*

*the event; in this perspective, it includes two false alarms that could be probably*

*deleted defining an updated curve; for example, are you sure that 20 days are the*

*proper time span for such events or is the K-coefficient adequate? A sensitivity*

*analysis could be produced.*

We have rewritten the conclusion part and added a special part "5.1 about the two
above points that did not trigger debris flows" to fully illustrate this problem (Line
502-509).
*20) L484: please use "suitable" for "suit"*

*This has been amended.*
*21) the variable "time" should be cited; that is, can you retrieve a decreasing*
*number of occurrences since the year of the earthquake going forward in time?*
*On the other hand, do the slopes experience a behavior at increasing resilience*
*since the earthquake occurrence? In few words, soon after the event much mate-*
*rial prone to sloping is retrievable and then the mass movement can be triggered*
*by less heavy precipitations while, subsequently higher values could be required.*
Yes, you are right. Actually, the rainfall threshold in the earthquake-hit areas would
increase after a certain time since the year of the earthquake. We have added a special
part to discuss this problem (Line 532-542).

**Response to Anonymous Referee #2:**
*1)General comments:*
*⋯In fact, the research developed by Zhuang et al. (2015) confirmed the decrease*

*of the rainfall influence with the increase of time interval, and established an*

*upper limit of 15 days. However, they found out that antecedent precipitation did*

*not significantly affect the soil water content, thus, its influence for the triggering*

*of debris flow was negligible. In fact, as several studies worldwide have demon-*

*strated, long-term antecedent precipitation is more likely to be related with*

*deep-seated landslides. In this sense, I would suggest a further discussion about*

*this issue as well as the uncertainties of the applied method.*

According to the previous studies, debris flow initiated is the result of the short dura-
tion rainfall (10-min rainfall, 1-h rainfall for example) and the effective antecedent
precipitations (Cui et al., 2007; Zhao, 2011; Guo, 2013; Zhuang, 2015). The precipi-
tation intensity is a measure of the peak precipitation. At the same time, the duration
of the peak precipitation is generally brief, lasting only up to tens of minutes. There-
fore, 10-min precipitation intensity (maximum precipitation over a 10-minute period
during the rainfall event) is selected as the stimulating rainfall for debris flow, which
is appropriate and most representative. However, it is difficult to obtain such
short-duration rainfall data in areas with scarcity of data which is just our research
range. Therefore, this study considered the effective antecedent precipitation of 20
days plus 1-h rainfall for the triggering of debris flow event. The significance role of
the antecedent rainfall has been further clarified in corresponding parts of the manu-
script, not only in "2.4 data collection and the characteristics of rainfall" (Line
253-258), "4.2.1 the calculation of the antecedent precipitation index (API)" (Line
433-441; Line 458-466), but also in the discussion section (Line 515-528).

*2) Finally, I suggest a rereading of the text to find some minor mistakes. Only as*

*an example: Page 4, line 98: assumptting;Page 24, line 481: mothodological*

*We have checked the whole manuscript thoroughly and amended the spelling errors.*

We wish that with the above revisions made, our manuscript can now be accepted for
publication on *Nature Hazards and Earth System Sciences* soon. Please do not hesi-
tate to contact me if you have any additional questions or comments.
Looking forward to hearing from you.
Regards
JIANG Yuanjun

[revised manuscript text omitted]

---

## Author Response (AR3)

Dear Samuele Segoni,

Thank you very much for everything you have done for us about our manuscript entitled "**Rainfall threshold calculation for debris flow early warning in areas with scarcity of data**" (No.nhess-2017-333). We truly appreciate all of the thoughtful comments from you and the Referee, and we have now revised our manuscript accordingly with a list of changes detailed below.

**Response to Referee 1#:**

*In general, all the arisen questions have been fulfilled. On my view, the choice of proxies could be not fully agreeable. About item 18), I confirm that, on my view, rainfall intensity should be reported on x-axis in terms of cumulative values over time unit (also if for 1 hour time reference, the values are the same). Moreover the readability of x-axis in Graphs forming Figure 13 could be improved; for example, reporting only hours and putting date as x-label.*

Thanks a lot for your suggestions and advices, and we fully considered your advices and redrew the figures in Figure 13.

**Response to Editor:**

*1)  I agree with the reviewer that the methodology has some major weaknesses. The choice of calculating API using 20 days is subjective and arbitrary. Why 20 and not 21, 30, 15, 10 or 5? Moreover, in several parts of the manuscript you state that in the literature other parameters are more frequently and conveniently used for debris flow modelling. Indeed, an overwhelming literature demonstrated that in similar cases of study I-D thresholds are more appropriate than thresholds accounting for antecedent rainfall conditions.*

Thank you very much for your kindly advices on this problem of this manuscript. As you suggested, firstly, we made 4 versions of the counting days when calculate API, 3days, 10days, 20days and 30days. The comparison among all the versions is shown in Table 6 (Line 469). It indicates that the value of the effective antecedent precipitations ($P_{a0}$) were increasing from 3 days to 20 days, while with the time last to 30 days, the value of $P_{a0}$ was barely changed anymore. Therefore, it can be considered that the effect of a rainfall event usually diminished in 20 days. Hence, the numbers of previous indirect rainfall days (n) is identified as 20. At the same time, we analyzed the trend of the trigger rainfall of debris flow events in Guojuanyan gully, and made a debris flow triggering thresholds for the gully (Line 485, Figure 14). The comparison result shows that the laws of the two threshold curve are the same and validates that the calculated method of rainfall threshold proposed in this work is reasonable. We also discussed this in the Discussions part of the manuscript(Line 488-497).

*2)  Fig. 7 and lines 234-260. This part is troublesome. You actually say that some kind of data are better than the ones you are going to use, then you state that you will use the worst data. You need to change this part because it introduces a major weakness in your research. Moreover, I do not agree with your conclusions. Fig.7 clearly shows that DF initiation cannot be characterized us-*

*ing only rainfall intensity, even if different time spans are taken into account (10',*
*1h and 1d intensity). You can get to this conclusion by observing that many DF*
*are below the annual average values. This encourages you to use antecedent*
*rainfall… HOWEVER, THESE PLOTS WOULD BE MORE INFORMATIVE IF*
*YOU PLOT DURATION ON THE X AXYS, to get I-D thresholds as mentioned in*
*the former comment (1b).*

We rewrote the whole paragraph about figure 7 to try to make it concise and clearly. As you suggested, the main conclusion of Fig.7 is that the trigger rainfall of the debris flow events had decreased obviously after the earthquake.

*3) The section 4.2.1 should be moved into the materials and methods section.*

*API should be explained the first time you mention it.*

This part has been moved to the materials and methods part as section 3.2 (Line 301-339).

*4) In the discussion section, you should clearly state that the threshold cannot be*

*applied elsewhere: the proposed approach is based on a procedure that can be*

*exported elsewhere only if a site-specific calibration is needed to develop specif-*

*ic thresholds for other test sites.*

This has been highlighted (Line 494-497).

*5) Others:*

We have checked the whole manuscript thoroughly again. Some additional and important references have also been added.

We wish that with the above revisions made, our manuscript can now be accepted for publication on *Nature Hazards and Earth System Sciences* soon. Please do not hesitate to contact me if you have any additional questions or comments.

Looking forward to hearing from you.

Regards

JIANG Yuanjun

[revised manuscript text omitted]